# LiteHall: A Three-Stage, Modular and Lightweight Pipeline for End-to-End Hallucination Detection

## Abstract

Large Language Models (LLMs) are increasingly applied in high-stakes domains such as medicine and law, where hallucinations can have serious consequences. Existing detection approaches either depend on costly proprietary LLMs with limited adaptability, or on monolithic open-source models that require full retraining, struggle with long evidence contexts, and lack transparency. We introduce LiteHall, a lightweight, fully open-source, three-stage hallucination detection pipeline designed for modularity, domain adaptability, and interpretability. Each stage leverages a 1.7B-parameter Small Language Model (SLM) trained independently with stage-specific Reinforcement Learning with Verifiable Rewards (RLVR) over a high-quality synthetic corpus of 120K+ examples, enabling efficient specialization without reliance on large monolithic models. To advance rigorous evaluation, we present HaFin500, a fine-grained benchmark of 500 long-form QA pairs spanning 30 fact-seeking domains, annotated with 6K claims and 3.5M evidence tokens. Extensive experiments show that LiteHall consistently surpasses both open-source and proprietary detectors. On out-of-domain benchmarks, LiteHall achieves substantial gains over strong baselines, including +6.4% / +10.0% (Accuracy/F1) against MiniCheck-7B, +6.1% / +4.8% over SAFE (GPT-3.5-turbo), +11.5% / +13.0% over AlignScore, and +9.8% / +15.2% over FAVA. Even compared to GPT-4o, LiteHall delivers +4.7% / +3.0% improvements in zero-shot mode, while retaining an additional +2.0% / +0.9% advantage when GPT-4o is integrated as a backbone. These results demonstrate that LiteHall not only matches or exceeds in-domain performance but also generalizes robustly out-of-domain, establishing it as a practical, transparent, and reproducible solution for trustworthy LLM deployments.

## 1 Introduction

Large Language Models (LLMs) such as GPT-4 and Llama (Brown et al., 2020; Maaten & the Llama Team, 2024) excel at generating fluent responses but frequently produce hallucinations—factually incorrect or unverifiable content (Huang et al., 2025; Tonmoy et al., 2024). This poses serious risks in high-stakes domains such as clinical diagnostics (Arvidsson et al., 2024), finance (Yang et al., 2024), and biomedical research (Lu et al., 2024), where factual reliability is paramount. While several hallucination detection methods have emerged (Gu et al., 2024; Wei et al., 2024; Chen et al., 2024), they often suffer from critical limitations: opaque decision processes (Sriramanan et al., 2024; Chen et al., 2024), reliance on proprietary LLMs (Wei et al., 2024; Li et al., 2024), and computationally intensive monolithic LLM backbones (>7B parameters) (Afzal et al., 2024; Wan et al., 2024; Yao et al., 2024). Furthermore, most offer only coarse-grained sentence-level predictions without exposing intermediate steps such as claim extraction or evidence retrieval, limiting transparency and adaptability.

In response, we ask:

> *Can hallucination detection be made lightweight, modular, and interpretable—without relying on large, proprietary models or compromising performance?*

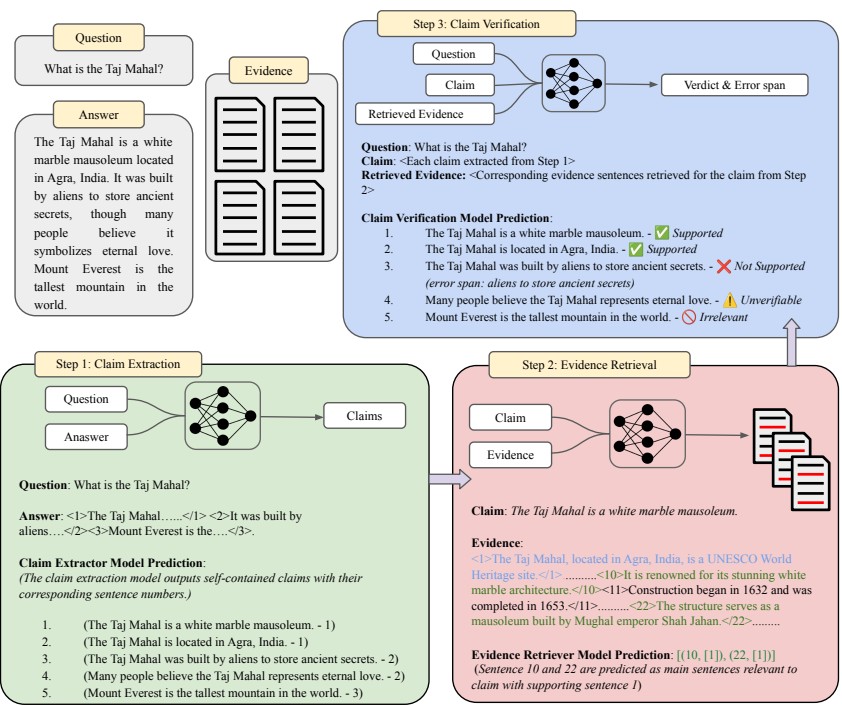

Figure 1: LiteHall pipeline: a three-stage hallucination detection framework for claim extraction, evidence retrieval, and claim verification.

We believe the answer lies in lightweight modular specialization. Our goal is to decompose the task into smaller, verifiable units that are optimized for both trustworthiness and efficiency. We introduce LiteHall, a modular hallucination detection pipeline built entirely using 1.7B-parameter Small Language Models (SLMs) (Li & Eldan, 2024; Lu et al., 2025). LiteHall divides the task into three stages—claim extraction, evidence retrieval, and verification—each trained independently using a two-stage regime: supervised fine-tuning (SFT) followed by reinforcement learning with verifiable rewards (RLVR) (DeepSeek-AI, 2025; Nayak et al., 2024; Sutton & Barto, 2018). This orchestration enables transparency, modular retraining, fine-grained control, and robust generalization (with RLVR)—key factors that are difficult to achieve in end-to-end monolithic LLM setups.

Recent work has shown that SLMs, when trained with verifiable signals and carefully structured supervision, can outperform larger models in reasoning and factuality tasks (Belcak et al., 2025; Luo et al., 2025; Dang & Ngo, 2025; Li et al., 2025). For example, small models have achieved >80% on AMC23 math tasks (Dang & Ngo, 2025), and RLVR-based approaches like LIMR (Li et al., 2025) demonstrate that training on carefully curated subsets can outperform brute-force data scaling. Yet, to our knowledge, no prior work applies RLVR-based training methods to hallucination detection in a modular, verifiable, and efficient manner.

To support fine-grained evaluation, we also release HaFin500, a benchmark of 500 examples spanning 30 topics with gold-standard claims, evidence, and hallucination spans. Extensive evaluation shows that LiteHall consistently outperforms state-of-the-art detectors—achieving average accuracy gains of +4.5% on ANAH-v2, +9.6% on KnowHalu, +6.4% on MiniCheck-7B, +11.5% on AlignScore, +9.8% on FAVA, and even surpassing GPT-4o zero-shot by +4.7%, despite using a fraction of its parameter size. These results validate that with well-orchestrated RLVR and task decomposition, smaller models can be highly precise. In summary, our key contributions are:

- A modular hallucination detection pipeline built on 1.7B-parameter SLMs with a module-wise RLVR design that orchestrates each component under verifiable reward signals for optimal performance.

- A new hallucination detection benchmark, HaFin500 with fine-grained annotations for claims, evidence, and hallucination spans across 30 domains.
- A fine-grained synthetic training corpus of 120K+ high-quality data points covering 30 fact-seeking domains, enabling both SFT and RLVR training of all modules on diverse, domain-grounded factual data.
- Extensive evaluation with ablations showing LiteHall's modular SLMs surpass prior methods and larger LLMs in hallucination detection.

## 2 Related Work

Early hallucination detection approaches treated the task as coarse binary classification—labeling entire responses as factual or hallucinated (Durmus et al., 2020; Dziri et al., 2022; Liu et al., 2022; Varshney et al., 2023; Manakul et al., 2023; Min et al., 2023). While efficient, these methods lacked diagnostic insights or localized attribution. Subsequent work introduced finer granularity, from claim-level detection (Ji et al., 2024a) to token-level spans. Internal methods such as LLM-CHECK (Sriramanan et al., 2024) and INSIDE (Chen et al., 2024) exploit model signals, while others like SELFCHECKGPT (Manakul et al., 2023) rely on cross-generation consistency. These reduce external dependencies but remain limited in adaptability and interpretability. Externally augmented methods strengthen factual grounding by querying evidence. SAFE (Wei et al., 2024) decomposes answers into atomic claims and reasons over retrieved snippets, but its reliance on proprietary APIs constrains reproducibility (Yao et al., 2024; Afzal et al., 2024; Wan et al., 2024). KNOWHALU (Zhang et al., 2025) introduces multi-step verification but depends on closed LLMs and lacks claim-level attribution. Annotation pipelines like ANAH-v2 (Gu et al., 2024) expand labeled datasets via iterative self-training, yet incur latency and semantic drift. Similarly, SELF-CHECKER (Li et al., 2024) supports plug-and-play verification but does not enable modular retraining. Among open approaches, LYNX (Ravi et al., 2024) surpasses OpenAI (2024) with a 70B model, though it remains resource-heavy and less interpretable. Furthermore, recent paradigms have explored dynamic, multi-agent frameworks for auditing the fact-checking capabilities of generalist LLMs, such as FACT-AUDIT (Lin et al., 2025). While highly effective for red-teaming conversational models via on-the-fly data generation, they function as dynamic evaluation tools rather than static, deployable detection pipelines. Because they rely on open-ended text justifications rather than structured, intermediate outputs (like exact sentence indices), they are fundamentally complementary to specialized, modular systems like LiteHall.

Overall, existing systems highlight trade-offs across scalability, transparency, and domain adaptability. We summarize these comparisons in Table 1, where LiteHall demonstrates advantages in interpretability, modularity, and efficiency.

Table 1: Comparison of hallucination detection methods. NoProp: avoids proprietary LMs; OpenLLM ($\leq$7B): uses open-source LMs $\leq$7B; LargeEvd: handles large evidence corpora; ModularTrainable: supports modular, independently trainable components; ClaimLevel: outputs claim-level verdicts; EvidenceCite & ErrorSpan: cites sentence-level evidence and flags hallucinated spans; Interpretability: offers interpretable explanations or spans; DomainAdapt: adapts to new domains without full retraining.

| Method | NoProp | OpenLLM ($\leq$7B) | Large Evidence | Modular Trainable | Claim-Level | Evidence Cite & Error Span | Domain Adapt |
|---|---|---|---|---|---|---|---|
| KnowHalu | ✗ | ✗ | ✗ | ✗ | ✗ | ✗ | ✗ |
| ANAH-v2 | ✓ | ✗ | ✓ | ✗ | ✗ | ✗ | ✓ |
| SAFE | ✗ | ✗ | ✓ | ✗ | ✓ | ✗ | ✗ |
| Lynx | ✓ | ✗ | ✗ | ✗ | ✗ | ✗ | ✓ |
| Self-Checker | ✗ | ✗ | ✓ | ✗ | ✓ | ✗ | ✗ |
| **LiteHall** | ✓ | ✓ | ✓ | ✓ | ✓ | ✓ | ✓ |

## 3 Methodology

### 3.1 LiteHall Pipeline Overview

*LiteHall* is a modular hallucination detection framework that decomposes the task into three lightweight, independently trainable stages: *Claim Extraction*, *Evidence Retrieval*, and *Claim Verification* (Figure 1). Each stage is powered by a fine-tuned 1.7B parameter model specialized for its subtask. Given an LLM answer $a$ (optionally with question $q$), LiteHall first extracts atomic, self-contained claims with sentence-level provenance (Section 3.2). For each claim, the retrieval stage gathers relevant evidence sentences from a large corpus, distinguishing *main* sentences that directly address the claim from *supporting* ones that provide contextual grounding (Section 3.3). These pairs are then passed to the verification stage, which assigns one of four factuality labels—`Supported`, `Not Supported`, `Unverifiable`, or `Irrelevant`—and also detect refuted spans (Section 3.4). By isolating subtasks and applying targeted fine-tuning, LiteHall achieves high factuality performance while minimizing computational overhead.

### 3.2 Claim Extraction Module

The claim extraction module identifies *self-contained factual claims* (Wei et al., 2024) from an LLM response, optionally paired with its query, and outputs atomic claims annotated with their originating sentence index (Step 1, Figure 1). Given an input $(q, a)$ or response $a$, the answer is decomposed into sentences $\{s_1, s_2, \ldots, s_n\}$, each tagged with indices for traceability:

$$\text{<1> } s_1 \text{ </1>, <2> } s_2 \text{ </2>, } \ldots, \text{ <n> } s_n \text{ </n>}$$

This schema supports sentence-level citation mapping and localized factuality scoring while preserving the reconstructed answer for downstream processing. A finetuned Qwen3-1.7B model (Yang et al., 2025), denoted $f_{\text{claim}}$, is then applied to the tagged response $a_{\text{tagged}}$ (and query $q$ if available) to extract factual claims with indices:

$$\{(z_i, j_i)\}_{i=1}^{m} = f_{\text{claim}}(q, a_{\text{tagged}})$$

where each $z_i$ is a self-contained claim and $j_i \in \{1, \ldots, n\}$ denotes its source sentence. The resulting claim set is thus $\mathcal{Z} = \{z_1, \ldots, z_m\}$, providing a structured, sentence-aligned representation of the LLM output.

### 3.3 Evidence Retrieval Module

The evidence retrieval module (Step 2 of LiteHall; Figure 1) identifies sentence-level evidence from a retrieved corpus to assess the veracity of extracted claims (Section 3.2). For each claim $z$, the retriever predicts a *main evidence sentence* and one or more *supporting sentences* that provide local context within candidate chunks.

**Input Preprocessing.** The retrieved corpus $\mathcal{C} = \{d_1, d_2, \ldots, d_K\}$ is divided into non-overlapping chunks of at most $T = 1500$ tokens (Zhao et al., 2024a), yielding $\mathcal{C}_{\text{chunked}} = \{c_1, \ldots, c_M\}$. Each chunk $c_j$ is split into sentences

$$c_j = \{s_{j1}, s_{j2}, \ldots, s_{jN_j}\},$$

and converted into a numbered sequence $\hat{c}_j$ for index-level traceability:

$$\hat{c}_j = \text{<1> } s_{j1} \text{ </1> <2> } s_{j2} \text{ </2>} \ldots \text{<}N_j\text{> } s_{jN_j} \text{ </}N_j\text{>}.$$

**Index Prediction.** Given a claim $z$ and chunk $\hat{c}_j$, a finetuned Qwen3-1.7B (Yang et al., 2025) retriever $f_{\text{ret}}$ predicts evidence indices:

$$\hat{y}_j^{(z)} = f_{\text{ret}}(z, \hat{c}_j) \quad \text{with} \quad \hat{y}_j^{(z)} = \left\{\left(m_j^{(i)}, \mathcal{S}_j^{(i)}\right)\right\}_{i=1}^{L_j},$$

where $m_j^{(i)}$ denotes the main sentence index and $\mathcal{S}_j^{(i)}$ the supporting indices. This ensures fidelity by explicitly linking claims to indexed source sentences.

**Evidence Reconstruction and Aggregation.** Indices are back-mapped to recover textual evidence groups:

$$\left(s_{jm_j^{(i)}}, \{s_{jk} \mid k \in \mathcal{S}_j^{(i)}\}\right),$$

yielding coherent passages that combine the main and supporting sentences. Aggregating across all chunks produces the complete evidence set:

$$\mathcal{E}^{(z)} = \bigcup_{j=1}^{M} \bigcup_{i=1}^{L_j} \left(s_{jm_j^{(i)}}, \{s_{jk} \mid k \in \mathcal{S}_j^{(i)}\}\right),$$

which is forwarded to the claim verification module (Section 3.4).

**Design Rationale.** Predicting both main and supporting indices instead of generating spans preserves the original document phrasing, guarantees citation traceability, reduces inference latency, and scales effectively to long-context retrieval—making it well-suited for LiteHall's modular pipeline.

## 3.4 Claim Verification Module

The final stage of LiteHall (Step 3 in Figure 1) verifies claims extracted in Section 3.2 against retrieved evidence (Section 3.3). Each claim is labeled and aggregated into sentence- and answer-level factuality scores, yielding a binary hallucination verdict.

**Formulation.** For claim $z_i \in \mathcal{Z}$ with evidence set $\mathcal{E}^{(z_i)}$, the verifier input is $x_i = (q, z_i, \mathcal{E}^{(z_i)})$. A finetuned Qwen3-1.7B (Yang et al., 2025) model $f_\theta^{\text{verify}}$ predicts

$$f_\theta^{\text{verify}}(x_i) = \left(y_i, \mathcal{R}_i \text{ if } y_i = \texttt{Not Supported}\right),$$

where $y_i \in \{\texttt{Supported}, \texttt{Not Supported}, \texttt{Unverifiable}, \texttt{Irrelevant}\}$. Here, $\mathcal{R}_i$ marks refuted spans. Labels denote: `Supported` (evidence aligns), `Not Supported` (contradiction), `Unverifiable` (insufficient/subjective), and `Irrelevant` (no contextual link) (Thorne et al., 2018; Wei et al., 2024; Li et al., 2024).

**Scoring and Verdict.** Given a set of claims $\mathcal{Z} = \{z_1, \ldots, z_N\}$ for an answer $a$, we partition them based on their predicted labels: $\mathcal{Z}_{\text{factual}} = \{z_i \mid y_i = \texttt{Supported}\}$, and $\mathcal{Z}_{\text{hallucinated}} = \mathcal{Z} \setminus \mathcal{Z}_{\text{factual}}$.

The module then computes a global FactScore (Min et al., 2023) for total candidate answer:

$$\text{FactScore}(a) = \frac{|\mathcal{Z}_{\text{factual}}|}{|\mathcal{Z}|}$$

To derive sentence-level insights, we associate each sentence $s_j$ with a subset of claims $\mathcal{Z}_j \subseteq \mathcal{Z}$, as annotated in Section 3.2. The corresponding sentence-level score is:

$$\text{FactScore}(s_j) = \frac{|\mathcal{Z}_{j,\text{factual}}|}{|\mathcal{Z}_j|}, \quad \text{where } \mathcal{Z}_{j,\text{factual}} = \mathcal{Z}_j \cap \mathcal{Z}_{\text{factual}}$$

An answer is `non-hallucinated` if $\text{FactScore}(a) \geq 0.75$, and `hallucinated` otherwise. The threshold (0.75) was tuned on held-out dev sets and validated across benchmarks (Koyejo et al., 2014), though adjustable per domain.

## 3.5 Training Details.

All three modules in LiteHall were trained using a two-stage process combining SFT and reinforcement learning optimization. The *claim extractor* was trained via SFT on 5,000 structured synthetic examples (Busker et al., 2025), followed by Direct Preference Optimization (DPO) (Rafailov et al., 2023) on 8,300 GPT-4o-generated pairwise comparisons. The synthetic training dataset and training procedure are described in Appendix A.2 and Appendix B.1, respectively. The *evidence retriever* underwent SFT on 25,000 synthetic samples and was further refined using Group Relative Policy Optimization (GRPO) (Shao et al., 2024; DeepSeek-AI,

2025) on 71,000 examples, guided by rewards for factuality and format adherence. Details on training data construction and training are in Appendix A.3 and Appendix B.2. Finally, the *claim verifier* was SFT-trained on 15,000 synthetic examples, then GRPO-optimized on 41,000 more using composite rewards targeting both verdict correctness and span-level hallucination detection. See Appendix A.4 for dataset creation and Appendix B.3 for training methodology. All synthetic data were generated via structured prompting pipelines using GPT-4o (see Appendix A).

### 3.6 HaFin500 Benchmark

While prior benchmarks such as FEVER (Thorne et al., 2018), HALUEVAL (Li et al., 2023), FELM (Chen et al., 2023), and FACTSCORE (Min et al., 2023) advanced hallucination detection, they mostly emphasize final verdict labels on short responses and lack supervision for intermediate steps like claim segmentation, evidence grounding, and error localization (Mishra et al., 2024). A broader comparison is provided in Table 13, Appendix D. To address this gap, we introduce HAFIN500, a benchmark covering every stage of hallucination detection. It spans 30 fact-seeking topics across healthcare, science, policy, and education, with an average of 17 questions per topic and long-form (Wei et al., 2024) GPT-4o answers, yielding ∼500 QA pairs. Each is paired with topic-specific references forming ∼7,000-token evidence contexts, producing ∼6,000 claims grounded in a 3.5M-token corpus. Claims are annotated with answer sentence origins, supporting/contextual evidence (with index + raw text), factuality labels, and localized hallucination spans. Annotation employed a multi-LLM voting scheme—GPT-4o (OpenAI, 2024), Claude-3.5-Sonnet (Anthropic, 2024), Gemini-2.5-Flash (DeepMind, 2025)—plus a rigorous human evaluation study on a stratified subset of 100 instances. This study confirmed high inter-annotator agreement (Fleiss' $\kappa = 0.69$–$0.78$) and a strict human-to-benchmark agreement of up to 91.0% for verdict labels, ensuring high annotation fidelity, realistic error injection, and minimal model bias. Full details of the human validation methodology and metrics are provided in Appendix D.5 HAFIN500 thus enables end-to-end evaluation: claim extraction, evidence retrieval, verdict prediction, and fine-grained error attribution. Full protocols, prompts, and validation details appear in Appendix D.

## 4 Experiments

### 4.1 Experimental Setup

**Evaluation Datasets** We evaluate LiteHall on both in-domain and out-of-domain settings. The in-domain suite includes five held-out test splits from the corresponding dataset: *HaluEval-QA* (Li et al., 2023), *HaluEval-SUMM* (Li et al., 2023), *ANAH* (Ji et al., 2024b), *COVID-QA* (Möller et al., 2020), and *RAGTruth* (Niu et al., 2024).For out-of-domain generalization, we test on five diverse datasets spanning natural and synthetic hallucinations: *AggreFact* (Tang et al., 2023), *FAVA* (Mishra et al., 2024), *FactScore* (Min et al., 2023), *FEVER* (Thorne et al., 2018), and our proposed *HaFin500*.

**Baselines** We benchmark LiteHall against strong baselines for both in-domain and out-of-domain hallucination detection. For in-domain, we include ANAH-v2 (Gu et al., 2024) (structured 3-stage factuality classification), KnowHalu (Zhang et al., 2025) (modular pipeline with reasoning over mixed knowledge), Lynx (70B) (Ravi et al., 2024) (explanation-generating verdict model), and GPT-4o (LiteHall) (OpenAI, 2024) (LiteHall pipeline with GPT-4o replacing SLMs). For out-of-domain, we compare with MiniCheck-7B (Tang et al., 2024) (single-model binary detector), AlignScore (Zha et al., 2023) and MiniCheck-FT5 (Tang et al., 2024) (NLI-based detectors), SAFE (Wei et al., 2024) (LiteHall-style pipeline with both GPT-3.5 and GPT-4o), FAVA (Mishra et al., 2024) (fine-grained hallucination model with 7B backbone), GPT-4o (zero-shot) (OpenAI, 2024) (prompt-only hallucination detection), and GPT-4o (LiteHall) (OpenAI, 2024) (full LiteHall pipeline with GPT-4o modules) and LettuceDetect-large-v1 (Kovács & Recski, 2025) (a recent state-of-the-art token-level RAG hallucination detection architecture).This way, we compare LiteHall against a diverse set of baselines spanning model types, reasoning styles, and evaluation settings, and additionally, for each LiteHall stage in isolation, we select suitable baselines tailored to the corresponding module.

**Metrics** We assess LiteHall with both *end-to-end* and *module-level* metrics for a holistic evaluation. For the overall hallucination detection task, we report Accuracy and F1 Score (Sokolova & Lapalme, 2009) under

a standard classification setting. In module-level, Claim extraction is evaluated with a custom *Semantic Match F1*, which extends beyond exact matches by rewarding semantically equivalent paraphrases via cosine similarity of Sentence-BERT embeddings (Reimers & Gurevych, 2019) (Appendix C.1). For the evidence retrieval module, we introduce tailored retrieval metrics: Precision, Recall, and F1 Score (redefined for retrieval), along with Coverage (capturing whether all gold evidence is retrieved) and Jaccard Similarity (overlap between predicted and gold sets). These differ from standard classification metrics and were designed specifically for this task (Appendix C.2). Finally, claim verification is evaluated with standard classification metrics, Accuracy and F1 Score. To rigorously verify the statistical validity of our performance gains and address uncertainty, we report 95% Confidence Intervals (CIs) using non-parametric bootstrap resampling ($N = 10,000$ iterations) for both in-domain and out-of-domain benchmarks. Furthermore, we conduct statistical significance testing on accuracy improvements using McNemar's Test. To maintain clarity, we report significance tests on a selected set of key baseline-dataset combinations carefully chosen to cover diverse evaluation scenarios—such as comparing our modular SLM against a monolithic model (e.g., MiniCheck-7B), a zero-shot frontier LLM (e.g., GPT-4o), and a modular pipeline powered by a frontier LLM (GPT-4o LiteHall). A summary of these statistical evaluations is provided in Appendix F

### 4.2 Results

#### 4.2.1 In-Domain Evaluation

We assess end-to-end hallucination detection performance following the standard methodology adopted in prior benchmarks (Li et al., 2023; Thorne et al., 2018). Across five in-domain benchmarks, LiteHall consistently surpasses both specially fine-tuned hallucination detectors and detectors leveraging large proprietary LLMs, as shown in Table 2. On *ANAH*, LiteHall improves over ANAH-v2 by +2.1% accuracy, despite ANAH-v2 being a 7B monolithic detector trained on its own domain-specific data; notably, LiteHall also outperforms ANAH-v2 by +9.1% on *HaluEval-QA*, showcasing stronger cross-task generalization. Compared to KnowHalu, which leverages GPT-3.5-turbo in a modular pipeline, LiteHall yields an average +8.1% accuracy gain, illustrating the advantages of training SLMs under strong verifiable reward signals. Averaged across datasets, LiteHall attains a +1.1% accuracy improvement over GPT-4o (LiteHall variant), showing that an ensemble of RLVR-optimized SLMs, when combined with carefully curated synthetic training data, can rival—and even outperform—much larger models in hallucination detection. Even against Lynx (70B), LiteHall delivers average higher accuracy despite operating with only ∼5B combined parameters, underscoring the efficacy of modular decomposition and lightweight orchestration in hallucination detection. Statistical testing confirms that LiteHall's improvements over prior architectures are highly robust; for instance, the +16.2% and +10.7% accuracy gains over KnowHalu on HaluEval-QA and ANAH yield $p < 0.001$ (McNemar's test). While the margin between LiteHall and the GPT-4o-backed LiteHall pipeline on datasets like HaluEval-QA is statistically comparable ($p = 0.125$), it is a substantial achievement that a ∼5B parameter SLM ensemble achieves statistical parity with a massive proprietary LLM. Full 95% Confidence Intervals and significance tests are detailed in Appendix F.

#### 4.2.2 Out-of-Domain Evaluation

As shown in Table 3, LiteHall achieves consistent and substantial gains across five diverse OOD benchmarks, outperforming both lightweight NLI-based detectors and large proprietary LLMs. Against MiniCheck-FT5 and MiniCheck-7B, LiteHall improves by +8.5%/+11.7% and +6.4%/+10.0% (Acc/F1), respectively, owing to its ability to decompose interleaved claims, reason across large evidence sets, and perform multi-hop reasoning—capabilities that NLI-style classifiers with length and reasoning bottlenecks fundamentally lack. Compared to AlignScore, LiteHall delivers a +11.5%/+13.0% margin, highlighting the limitations of single-score factuality metrics when faced with long, complex responses. LiteHall also surpasses FAVA by +9.8%/+15.2%, as its modular RLVR-trained pipeline specializes each stage of extraction, retrieval, and verification, enabling richer evidence integration than FAVA's single-model approach. Against SAFE (GPT-3.5-turbo), LiteHall secures +6.1%/+4.8% gains, despite SAFE's stronger backbone, underscoring the advantage of carefully curated synthetic training data and specialization. Even compared to GPT-4o, LiteHall maintains an edge, improving by +4.7%/+3.0% in zero-shot settings and +2.0%/+0.9% when GPT-4o is used as a LiteHall module drop-in, demonstrating that the full pipeline outperforms generic prompting or single-

Table 2: End-to-End Hallucination Detection Performance on In-Domain Datasets

| Method | Metric | HaluEval-QA | HaluEval-SUMM | ANAH | COVID-QA | RAGTruth |
|---|---|---|---|---|---|---|
| ANAH-v2 | Acc | 80.8 | 66.4 | 89.6$^\star$ | 90.9 | 75.7 |
|  | F1 | 79.2 | 66.0 | 89.3$^\star$ | 88.7 | 66.5 |
| KnowHalu | Acc | 73.7 | 66.9 | 81.0 | 89.4 | 77.7 |
|  | F1 | 71.8 | 67.0 | 78.4 | 87.8 | 68.5 |
| Lynx (70B) | Acc | 88.4$^{\star\dagger}$ | N/A$^\dagger$ | N/A$^\dagger$ | 97.5$^{\star\dagger}$ | 80.2$^\dagger$ |
| GPT-4o(LiteHall) | Acc | 88.2 | 69.4$^\star$ | 87.4 | 96.1 | **83.0** |
|  | F1 | 86.5$^\star$ | 68.8$^\star$ | 87.0 | 95.4 | **71.8** |
| **LiteHall** | Acc | **89.9** | **70.0** | **91.7** | **97.6** | 80.2$^\star$ |
|  | F1 | **87.2** | **70.0** | **90.6** | **96.6** | 70.4$^\star$ |

$^\dagger$ Except for this row, all results in this table are reproduced by us. The numbers for Lynx (70B) are cited directly from the original paper because we do not have the computational budget to run a 70B model. "N/A" indicates metrics that were not available in the original paper.

model substitutes. These results collectively emphasize that LiteHall's modular architecture, strengthened by RLVR optimization and high-quality synthetic data, delivers robust generalization across domains where alternative detectors falter. To ensure these out-of-domain gains are not artifacts of dataset variance, we conducted paired significance testing across key baseline comparisons (see Appendix G). LiteHall's accuracy improvements over strong monolithic models like MiniCheck-7B on AggreFact ($p = 0.042$) and GPT-4o (zero-shot) on HaFin500 ($p = 0.002$) are statistically significant, confirming the consistent generalization capability of the RLVR-tuned modular architecture.

### 4.2.3 Efficiency–performance trade-offs

To make LiteHall's computational profile explicit, we report end-to-end latency and peak GPU memory in a unified inference setup (`fp16` precision, KV cache enabled, 2×A100 80GB) for LiteHall and strong open-source baselines in Table 4. Under this common setting, LiteHall achieves favorable efficiency–performance trade-offs against all 7B–70B detectors: compared to ANAH-v2-7B (sentence-level detection), LiteHall uses only $\approx 0.47\times$ the peak GPU memory and is $\approx 2\times$ faster, while improving Accuracy/F1 by $+7.9/ + 9.5$ on average across out-of-domain benchmarks; relative to FAVA-7B (token-level), LiteHall requires $\approx 0.55\times$ the memory and is $\approx 1.2\times$ faster, yet still yields $+9.8/ + 15.2$ higher Accuracy/F1. Against MiniCheck-7B (response-level), LiteHall's token-level detector uses $\approx 0.6\times$ of MiniCheck's peak memory and trades $\approx 1.9\times$ higher latency for substantially better factuality ($+6.4/ + 10.0$ Accuracy/F1), highlighting a controlled cost for much finer-grained judgments. Crucially, when compared to Lynx-70B (response-level), LiteHall attains higher Accuracy/F1 ($+4.2/ + 1.6$) while using only $\approx 0.08\times$ the peak memory ($\approx 13\times$ more memory-efficient) and being $\approx 3.2\times$ faster, demonstrating that a modular $\approx 5.1$B SLM ensemble can outperform a large 70B monolith in both quality and efficiency. We restrict Table 4 to open-source models where we can directly measure hardware-level telemetry; proprietary APIs such as GPT-4o are excluded due to the lack of reliable, controlled latency and memory measurements. Overall, these results show that LiteHall delivers strictly stronger factuality performance—at finer, token-level granularity—while using substantially less memory and competitive or better latency than monolithic 7B–70B detectors, supporting its practicality for both real-time and large-scale batch deployment.

### 4.2.4 Module-wise Performance (Third-party Benchmarks)

**Module-wise Performance.** To assess the robustness of each module in isolation, we stress-tested LiteHall's stages on public benchmarks aligned with their sub-tasks. For claim extraction (FactScore, Table 6), LiteHall's sentence-indexed extractor surpasses SAFE by $+5.1\%$ and GPT-4o by $+3.5\%$ in Semantic Match

Table 3: End-to-End Hallucination Detection Performance on Out-of-Domain Datasets.

| Method | Metric | AggreFact | FAVA | FactScore | HaFin500 | FEVER | Avg |
|---|---|---|---|---|---|---|---|
| MiniCheck-FT5 | Acc | 76.3 | 69.6 | 76.5 | 68.8 | 86.9 | 75.6 |
|  | F1 | 77.1 | 24.7 | 79.6 | 60.9 | 86.1 | 65.7 |
| MiniCheck-7b | Acc | 78.6 | 71.8 | 79.0 | 71.1 | 88.2 | 77.7 |
|  | F1 | 79.3* | 25.3 | 81.2 | 63.8 | 87.2 | 67.4 |
| Lettucedetect-large-v1 | Acc | 76.1 | 70.1 | 77.4 | 67.5 | 87.1 | 75.6 |
|  | F1 | 77.4 | 25.0 | 80.6 | 60.3 | 86.7 | 66.0 |
| AlignScore | Acc | 71.1 | 59.7 | 77.7 | 68.7 | 85.6 | 72.6 |
|  | F1 | 72.7 | 24.2 | 80.0 | 59.2 | 85.8 | 64.4 |
| FAVA | Acc | 73.5 | 74.4* | 69.9 | 73.8 | 79.9 | 74.3 |
|  | F1 | 75.9 | 44.0 | 63.9 | 53.0 | 74.3 | 62.2 |
| SAFE (gpt-3.5-turbo) | Acc | 76.9 | 67.9 | 76.7 | 78.3 | 90.1 | 78.0 |
|  | F1 | 76.2 | 44.1 | 75.5 | 77.9 | 89.4 | 72.6 |
| SAFE (gpt-4o) | Acc | 77.4 | 68.4 | 83.6 | 82.9 | 89.3 | 80.56 |
|  | F1 | 76.1 | 44.2 | 82.5 | 82.2 | 88.9 | 74.94 |
| GPT-4o (zero-shot) | Acc | 76.4 | 67.6 | 82.1 | 82.1 | 89.0 | 79.4 |
|  | F1 | 76.0 | 43.7 | 81.9 | 81.7 | 88.6 | 74.4 |
| GPT-4o (LiteHall) | Acc | 78.8* | 68.8 | **85.2** | 86.5* | **91.2** | 82.1* |
|  | F1 | 77.1 | 44.4* | **84.7** | 85.8* | **90.4** | 76.5* |
| **LiteHall** | Acc | **80.7** | **76.7** | 84.1* | **88.7** | 90.3* | **84.1** |
|  | F1 | **81.6** | **45.5** | 83.9* | **86.1** | 89.7* | **77.4** |

Table 4: Peak GPU memory, average latency, and hallucination detection performance (accuracy and F1) across OOD datasets from Table 3.

| Method | Model size | Hallucination detection level | Accuracy | F1 | Avg. latency (s) | Peak GPU memory (GiB) |
|---|---|---|---|---|---|---|
| MiniCheck-7b | 7b | Response | 77.7 | 67.4 | 1.4 | 20.3 |
| FAVA | 7b | Token | 74.3 | 62.2 | 3.1 | 21.4 |
| ANAH-v2 | 7b | Sentence | 76.2 | 67.9 | 5.1 | 24.8 |
| Lynx (70B) | 70b | Response | 79.9 | 75.8 | 8.2 | 151.7 |
| **LiteHall** | 5.1b | Token | **84.1** | **77.4** | **2.6** | **11.72** |

F1 (C.1), where explicit sentence tags and atomic decoding help mitigate span drift. In evidence retrieval (FEVER, Table 8), LiteHall outperforms SuperPAL (Ernst et al., 2021) by +11.4% F1 and +12.2% Coverage, and RoBERTa-AS2 (Di Liello et al., 2022) by +10.4% F1 and +12.7% Coverage, while remaining competitive with GPT-4o (−2.1% F1, +0.7% Coverage). These retrieval-specific metrics (C.2) highlight LiteHall's ability to gather concise, high-quality evidence while tolerating minimal noise. Finally, in claim verification (HaluEval-QA, Table 5), LiteHall achieves substantial gains over AlignScore (+12.3% F1) and MiniCheck-7B

(+8.2% F1), and nearly matches GPT-4o (−0.1% F1), showing that focused evidence coupled with lightweight reasoning yields verdict quality on par with state-of-the-art large models.

Table 5: Claim Verification Performance

| Metric / Model | F1 | Accuracy |
|---|---|---|
| AlignScore | 74.1 | 79.3 |
| MiniCheck-FT5 | 75.6 | 80.5 |
| MiniCheck-7b | 78.2 | 82.6 |
| GPT-4o | **86.5** | 88.2* |
| **LiteHall (Claim Verifier)** | 86.4* | **89.0** |

Table 6: Claim Extraction Performance

| Metric / Model | F1 |
|---|---|
| SAFE (gpt-3.5-turbo) | 61.7 |
| GPT-4o | 63.3* |
| **LiteHall (Claim Extractor)** | **66.8** |

**Span-Level Evaluation.** To assess LiteHall's fine-grained hallucination detection, we evaluate its claim extractor on *FavaBench* using span-level metrics across six hallucination error types, as summarized in Table 7. LiteHall surpasses FAVA with an average gain of +1.8% accuracy and +1.6% F1, with particularly strong improvements on challenging categories such as *Contradictory* (+3.9% accuracy) and *Entity* (+10.6% accuracy). These results highlight LiteHall's effectiveness in precisely identifying semantically inconsistent spans.Since LiteHall outputs spans without type labels, we use OpenAI o3 to assign error categories under FAVA's rubric, enabling fair, type-consistent scoring and highlighting LiteHall's strength in localized, interpretable hallucination detection.

Table 7: LiteHall Span-Level Evaluation (Acc/F1) by Error Type

| Method | Entity | Relation | Contradictory | Invented | Subjective | Unverifiable | Average |
|---|---|---|---|---|---|---|---|
| FAVA | 63.3 / 54.7 | **81.2 / 35.1** | 62.6 / **37.8** | 76.7 / 24.7 | 79.6 / 43.0 | **78.7 / 40.8** | 73.7 / 39.4 |
| **LiteHall** | **73.9 / 62.9** | 73.5 / 34.4 | **66.5** / 28.8 | **82.2 / 38.0** | **82.8 / 44.3** | 74.1 / 37.7 | **75.5 / 41.0** |

#### 4.2.5 Ablation Studies

**Contribution of Individual Modules.** To further validate LiteHall's modular design, we ablate core components and observe performance degradation (Table 9). Removing the *Claim Extractor*, which forces the verifier to label entire responses without atomic segmentation, leads to sharp drops (–6.8%/–5.5% Acc/F1 on HaFin500 and –3.8%/–4.2% on FavaBench), underscoring the necessity of localized claim identification to capture span-level inconsistencies. Similarly, bypassing the *Evidence Retriever* and supplying full corpus chunks instead of targeted sentences degrades accuracy and F1 by –5.2%/–4.2% and –2.9%/–3.1%, respectively, confirming the benefit of precise sentence-level filtering over noisy full-context inputs. Together, these ablations validate that LiteHall's effectiveness hinges on modular decomposition, where each stage materially contributes to overall robustness.

Table 8: Evidence Retrieval Performance

| Metric / Model | Coverage | Precision | Recall | F1 | Jaccard |
|---|---|---|---|---|---|
| SuperPAL | 71.0 | 61.4 | 71.3 | 66.0 | 49.2 |
| RoBERTa-AS2 | 70.5 | 63.8 | 70.5 | 67.0 | 50.3 |
| GPT-4o | 80.6* | **73.2** | 80.6* | 76.7* | 62.2* |
| **LiteHall (Evidence Extractor)** | **83.2** | 71.1* | **84.9** | **77.4** | **63.1** |

Table 9: Contribution of each LiteHall Modules on HaFin500 and FavaBench (Acc / F1)

| Model Variant | HaFin500 | FavaBench |
|---|---|---|
| LiteHall (full) | **88.7 / 86.1** | **76.7 / 45.5** |
| w/o Claim Extractor | 81.9 / 80.6 | 72.9 / 41.3 |
| Δ vs. full | -6.8 / -5.5 | -3.8 / -4.2 |
| w/o Evidence Retriever | 83.5 / 81.9 | 73.8 / 42.4 |
| Δ vs. full | -5.2 / -4.2 | -2.9 / -3.1 |

**LiteHall's Benefit Beyond Training Data.**  To disentangle the role of training data from architectural and RLVR optimization choices, we trained a strong monolithic baseline—MiniCheck-7B—on the exact synthetic data used by LiteHall. As shown in Table 10, this up-training yields gains of +3.9% Accuracy and +2.9% F1 over the original MiniCheck-7B, confirming the usefulness of the data itself. However, LiteHall still exceeds this retrained baseline by +6.0% Accuracy and +5.2% F1 on average, underscoring that its advantage derives not only from data but from its modular stage-wise decomposition, targeted supervision, and RLVR-driven alignment of each module to its specific subtask.

Table 10: Comparing LiteHall vs. Monolithic 7B Model Trained on Same Data (Acc / F1)

| Method | HaluEval-QA | HaluEval-SUMM | ANAH | COVID-QA | RAGTruth | Avg |
|---|---|---|---|---|---|---|
| MiniCheck-7B (pretrained) | 76.0 / 71.4 | 61.5 / 64.8 | 78.3 / 81.3 | 90.1 / 90.4 | 74.1 / 66.5 | 76.0 / 74.9 |
| MiniCheck-7B (+LiteHall data) | 80.1 / 74.7 | 65.5 / 66.7 | 82.8 / 84.0 | 93.6 / 94.3 | 77.4 / 69.2 | 79.9 / 77.8 |
| LiteHall | **89.9 / 87.2** | **70.0 / 70.0** | **91.7 / 90.6** | **97.6 / 96.6** | **80.2 / 70.4** | **85.9 / 83.0** |

**Disentangling Pipeline and Model Contributions.**  To disentangle the contributions of LiteHall's modular architecture and the specialized SLMs, we conduct model-substitution ablations on the out-of-domain benchmarks (Table 3). First, replacing GPT-4o in a zero-shot setting with GPT-4o under the LiteHall pipeline yields a +2.7% Accuracy and +2.1% F1 improvement. This demonstrates that the modular decomposition and stage-specific supervision alone—independent of model size—substantially enhance robustness in hallucination detection. Next, substituting GPT-4o in the LiteHall pipeline with our trained 1.7B SLMs delivers an additional +2.0% Accuracy and +0.9% F1 gain. These results highlight that compact, SFT+RLVR-optimized SLMs not only rival but can surpass GPT-4o within the same pipeline, offering domain adaptability, interpretability, and efficiency while maintaining competitive accuracy.

**Effect of RLVR.**  To quantify the effect of RLVR, we compared LiteHall (SFT-only) to LiteHall (SFT + RLVR) on five out-of-domain benchmarks—FactScore, FEVER, HaFin500, AggreFact, and FavaBench (Table 11). Averaged over these datasets, RLVR delivers a clear and robust gain, improving performance from 79.4 → 84.1 Acc and 72.1 → 77.4 F1 (+4.7 Acc / +5.3 F1). The improvements are consistent across all benchmarks, with especially pronounced gains on long-context, reasoning-heavy settings such as FactScore (+6.4 Acc / +6.9 F1) and HaFin500 (+6.8 Acc / +7.1 F1), and non-trivial boosts on AggreFact (+4.8 / +3.9) and FavaBench (+2.8 / +2.1). These results indicate that RLVR is not a marginal refinement but a key driver of LiteHall's out-of-domain robustness, systematically strengthening both accuracy and F1 beyond what supervised fine-tuning alone can achieve.

# 5 Limitations and Future Work

LiteHall establishes that lightweight, modular architectures can surpass large proprietary systems in hallucination detection, while remaining transparent, adaptable, and cost-efficient. Its stage-wise decomposition enables targeted supervision, interpretability, and domain specialization, with experiments confirming robustness across both in-domain and out-of-domain benchmarks. Future directions include advancing module synergy through joint optimization strategies, exploring compact reasoning-enhanced models for deeper

Table 11: Effect of RLVR on LiteHall performance

| Method | Metric | Fact Score | FEVER | HaFin 500 | Aggre Fact | Fava Bench | Avg |
|---|---|---|---|---|---|---|---|
| LiteHall (SFT) | Acc | 77.7 | 87.4 | 81.9 | 75.9 | 73.9 | 79.4 |
| | F1 | 77.0 | 85.6 | 79.0 | 75.7 | 43.4 | 72.1 |
| LiteHall (SFT + RLVR) | Acc | 84.1 | 90.3 | 88.7 | 80.7 | 76.7 | 84.1 |
| | F1 | 83.9 | 89.7 | 86.1 | 81.6 | 45.5 | 77.4 |

factual inference, and extending LiteHall to support multimodal and knowledge-graph–driven retrieval. These avenues will further strengthen its scalability and reliability, positioning LiteHall as a practical and extensible foundation for trustworthy fact verification in real-world deployments. Despite its efficiency and modularity, LiteHall's sequential three-stage architecture is inherently susceptible to cascading errors, where inaccuracies in initial claim extraction can negatively impact downstream evidence retrieval and verification. Furthermore, the pipeline currently focuses on text-only processing and relies on an externally provided evidence corpus, limiting its immediate applicability in multimodal scenarios or dynamic, web-scale retrieval. Finally, while the specialized 1.7B-parameter models excel at targeted tasks, they may still encounter capacity bottlenecks when evaluating claims that require highly complex logical or mathematical reasoning, highlighting an avenue for future integration with stronger reasoning-enhanced backbones.

# 6 Reproducibility Statement

We release our training and test datasets, along with complete training and evaluation code, in the supplementary materials. Comprehensive experimental details—including methodology, evaluation metrics, and all hyper-parameters—are provided in the Appendix.

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

## A  Synthetic Training Data Generation Pipeline

To enable robust supervision across all modules of the LITEHALL pipeline, we constructed a high-quality synthetic training dataset through a multi-stage generation process. This process leverages large language models, specifically GPT-4o, and aligns closely with the modular structure of LITEHALL. The dataset preparation spans claim extraction, evidence retrieval, and claim verification, ensuring consistency and alignment between stages.

### A.1  Source Datasets

The synthetic data generation process began by sampling from six diverse datasets, including both publicly available and internally curated resources. The selected sources encompass a range of factuality tasks across QA and summarization settings. The sampling distribution across datasets is summarized below:

| Dataset | Samples |
|---|---|
| ANAH (Ji et al., 2024b) | 800 |
| COVID-QA (Möller et al., 2020) | 1,500 |
| HaluEval-QA (Li et al., 2023) | 2,000 |
| HaluEval-SUMM (Li et al., 2023) | 2,000 |
| RAGTruth (Niu et al., 2024) | 2,000 |
| Ours (long-form QA) | 5,000 |
| **Total** | **13,300** |

Each example in these datasets contains a factual answer, with many also including a corresponding question, gold-standard evidence passages, and, in some cases, human-annotated factuality labels. However, these datasets fall short in representing scenarios with long-form answers and extended evidence passages, particularly in settings requiring reasoning across large contexts. To address this, we augmented our pool with 5,000 long-form QA examples that include comprehensive evidence contexts to ensure sufficient coverage of document-level fact verification challenges. To ensure strict out-of-domain (OOD) generalization, the 5,000 synthetic long-form QA examples were generated by exclusively sampling topics from the in-domain splits (ANAH, COVID-QA, HaluEval-QA, HaluEval-SUMM, and RAGTruth). No novel OOD topics were introduced during this generation phase.

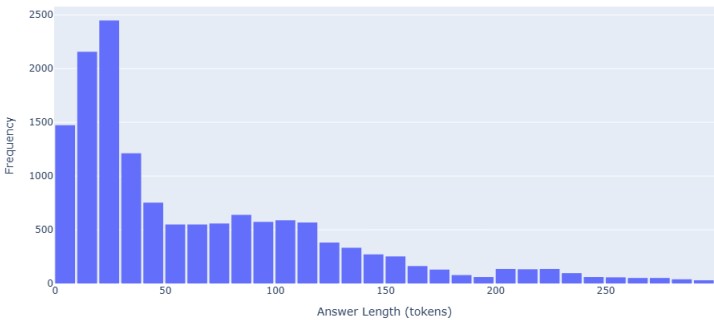

Figure 2: Distribution of Answer Lengths

## A.2 Stage 1: Claim Extraction Data Generation

The first step in our training data pipeline involved preparing supervision for the claim extraction module. For this, we provided GPT-4o with either a question-answer pair or an unprompted answer as input. Using the structured prompt detailed in Table 21, the model produced a list of self-contained factual claims. Each extracted claim was annotated with the sentence index from which it originated, enabling traceability and fine-grained supervision. This approach ensured that the model captured atomic factual units with precise contextual grounding.

The resulting dataset consisted of 13,300 QA or answer-only inputs, from which a total of 56,017 claims were extracted. Among these, 8,204 examples contained both a question and an answer. The average answer length across the dataset was 69 tokens. The distribution of answer lengths is visualized in Figure 2.

The claims generated in this stage served as direct input to the next phase—evidence retrieval—where the model learns to localize supporting context for each extracted claim.

## A.3 Stage 2: Evidence Retrieval Data Generation

Building upon the output of the claim extraction stage, we prepared data for the evidence retrieval module. For each extracted claim, we located the relevant gold evidence sentences from the corresponding dataset example. These evidence passages were segmented into non-overlapping chunks, each containing up to 1500 tokens. Within each chunk, sentences were indexed using structured tags to facilitate later reference.

Each claim was then paired with its associated evidence chunks to form multiple retrieval instances. These pairs were passed to GPT-4o along with the prompt in Table 22 to identify the most relevant sentence within the chunk (main sentence) and its supporting sentences. These sentence indices served as the supervisory signal for the evidence retriever model.

This step resulted in 96,017 claim-chunk training pairs, with each claim typically linked to approximately two main sentences per chunk. Each main sentence, in turn, was supported by around two additional sentences on average. The distribution of evidence sizes across the dataset is shown in Figure 3.

The retrieved sentence indices and their corresponding texts were then aggregated to form complete evidence contexts, which were used as input to the final verification stage.

## A.4 Stage 3: Claim Verification Data Generation

With both the claims (from Stage 1, A.2) and corresponding evidence contexts (from Stage 2, A.3) prepared, we constructed supervision data for training the claim verification module. For each claim, we aggregated all main and supporting sentences retrieved across the evidence chunks to form a comprehensive context. The resulting input consisted of the question (if available), the claim, and the consolidated evidence.

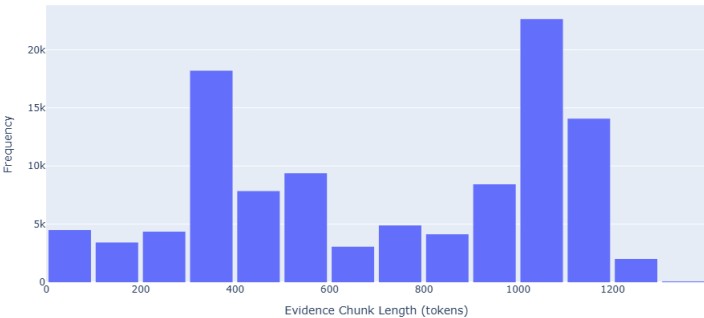

Figure 3: Distribution of Evidence Lengths

Using the prompt described in Table 23, GPT-4o was tasked with assigning one of four factuality labels—`Supported`, `Not Supported`, `Unverifiable`, or `Irrelevant`—to each claim. When the label was `Not Supported`, GPT-4o was additionally prompted to identify the specific error spans within the claim text. This provided token-level supervision to guide the model's hallucination detection capability.

The final dataset for claim verification consisted of 56,017 labeled claims, distributed as follows:

| Label | Count |
|---|---|
| Supported | 31,351 |
| Not Supported | 19,051 |
| Unverifiable | 3,551 |
| Irrelevant | 2,064 |

### A.5 Motivation for Synthetic Augmentation

While the original datasets provided valuable benchmarks, they were limited in two critical dimensions: the length of evidence required for verification and the complexity of long-form QA settings. Many examples in existing benchmarks rely on short, self-contained evidence snippets that do not reflect the real-world challenges of validating factuality in extended textual contexts. Moreover, the majority of QA-style datasets emphasize concise question-answering tasks that lack document-level dependencies.

To address these gaps, we constructed an additional set of 5,000 long-form QA examples. These were curated to include extended answers and their corresponding long evidence passages, ensuring that our pipeline could effectively learn to retrieve and verify factual content in realistic, high-context scenarios.

### A.5.1 Topic-Level Deduplication and Leakage Prevention

To guarantee zero train-test overlap between the synthetic training corpus and all test sets (particularly HaFin500), we implemented an automated hierarchical deduplication pipeline. Using Qwen3-8B, we extracted a granular 2- to 3-level topical taxonomy (e.g., $Healthcare \rightarrow Diseases \rightarrow Diabetes$) for every instance in the training and test sets. Cross-referencing these hierarchies confirmed a 0% overlap at the 2nd and 3rd taxonomy levels between the training data (including the 5,000 synthetic examples) and HaFin500, as well as all other OOD benchmarks. LiteHall's performance on HaFin500 is attributable to its architectural alignment with the benchmark's multi-stage, long-context design, rather than data contamination.

# B  Training Methodology

## B.1  Claim Extraction Model

### B.1.1  Notation and Data Splits

The pre-trained backbone is denoted as $\pi_{\theta_0}$ and corresponds to Qwen3-1.7B. A candidate input is either a question-answer pair or an unprompted answer, represented as $x$, with its corresponding gold claim list denoted $y^\star$.

- SFT Dataset: $\mathcal{D}_{\mathrm{SFT}} = \{(x_i, y_i^\star)\}_{i=1}^{5000}$
- DPO Dataset: $\mathcal{D}_{\mathrm{DPO}} = \{(x_j, y_j^+, y_j^-)\}_{j=1}^{8300}$, where:
    - $y_j^+$: claim set generated by GPT-4o (preferred)
    - $y_j^-$: claim set generated by base model $\pi_{\theta_0}$ (less preferred)

All stages use full-parameter fine-tuning via the `LLaMA-Factory` framework.

### B.1.2  Stage 1 – SFT

We minimize the negative log-likelihood over gold-standard claims:

$$\mathcal{L}_{\mathrm{SFT}}(\theta) = -\frac{1}{|\mathcal{D}_{\mathrm{SFT}}|} \sum_{(x, y^\star) \in \mathcal{D}_{\mathrm{SFT}}} \log \pi_\theta(y^\star \mid x)$$

- Learning Rate: $1 \times 10^{-5}$
- Global Batch Size: 16
- Epochs: 1
- Optimizer: AdamW
- Schedule: cosine decay

The resulting weights $\theta_{\mathrm{SFT}}$ initialize the second-stage policy and serve as the reference model in DPO.

### B.1.3  Stage 2 – DPO

For every triplet $(x, y^+, y^-)$ in the dataset, the DPO objective encourages higher preference for the GPT-4o-generated sample:

$$\Delta \log \pi_\theta = \log \pi_\theta(y^+ \mid x) - \log \pi_\theta(y^- \mid x), \quad \Delta \log \pi_{\mathrm{ref}} = \log \pi_{\theta_{\mathrm{SFT}}}(y^+ \mid x) - \log \pi_{\theta_{\mathrm{SFT}}}(y^- \mid x)$$

$$\mathcal{L}_{\mathrm{DPO}}(\theta) = -\frac{1}{|\mathcal{D}_{\mathrm{DPO}}|} \sum_{(x, y^+, y^-)} \log \sigma \left[ \beta \left( \Delta \log \pi_\theta - \Delta \log \pi_{\mathrm{ref}} \right) \right]$$

where $\sigma(z) = \frac{1}{1+e^{-z}}$, and $\beta = 0.1$ controls the sharpness of preference.

- Learning Rate: $1 \times 10^{-6}$
- Global Batch Size: 8
- Epochs: 1
- Loss Type: sigmoid

### B.1.4 Optimization Summary

$$\theta_0 \xrightarrow[\text{SFT, 1 epoch}]{\mathcal{L}_{\text{SFT}}, \text{1e-5}} \theta_{\text{SFT}} \xrightarrow[\text{DPO, 1 epoch}]{\mathcal{L}_{\text{DPO}}, \text{1e-6}} \theta^\star$$

### B.1.5 Implementation Details

- Framework: `LLaMA-Factory v0.9.2` (Zheng et al., 2024)

- Precision: Full-precision training; gradient checkpointing enabled

- Stability Measures: Gradient clipping at 1.0, dropout rate 0.0, RMSNorm e $= 1 \times 10^{-5}$

- Hardware: Training was conducted on 2×A6000 GPUs (48GB each).

### B.1.6 Rationale for Two-Stage Training

SFT provides a strong initialization by directly aligning the model with high-quality, human-annotated claims, enabling accurate and structured extraction. DPO further refines the model by contrasting preferred (GPT-4o) and less preferred (base model) outputs, encouraging more precise, self-contained, and contextually faithful claims. Together, this two-stage training ensures both fidelity to annotated supervision and preference alignment with stronger generative behaviors.

## B.2 Evidence Retrieval Model

The training of the sentence-index-based evidence retriever $\pi_\theta$ is conducted in two sequential phases: a SFT phase for grounding, followed by a reinforcement learning phase using GRPO for reward-aligned retrieval.

### B.2.1 Stage 1 — SFT

**Model and Initialization.** We initialize $\pi_\theta$ from the base language model QWEN3-1.7B, targeting sentence-level evidence index prediction for each input.

**Training Dataset.** We define the SFT training dataset as:

$$\mathcal{D}_{\text{SFT}} = \{(x^{(n)}, y^{(n)})\}_{n=1}^{25,000},$$

where each input $x^{(n)} = (z^{(n)}, \hat{c}^{(n)})$ consists of a retrieved passage chunk $z^{(n)}$ and a claim $\hat{c}^{(n)}$, and $y^{(n)}$ contains the gold indices for main and supporting evidence.

**Objective Function.** The supervised training minimizes the negative log-likelihood over $\mathcal{D}_{\text{SFT}}$:

$$\mathcal{L}_{\text{SFT}}(\theta) = -\frac{1}{|\mathcal{D}_{\text{SFT}}|} \sum_{n=1}^{|\mathcal{D}_{\text{SFT}}|} \log \pi_\theta(y^{(n)} \mid x^{(n)}).$$

**Optimization.** Training is performed for one epoch using the ADAMW optimizer with a learning rate of $1 \times 10^{-5}$ and a global batch size of 16. This results in model parameters $\theta_{\text{SFT}}$.

### B.2.2 Stage 2 — GRPO

**Unlabeled Pool.** A larger, unlabeled dataset of 71,000 claim–chunk pairs is defined as:

$$\mathcal{D}_{\text{GRPO}} = \{x^{(m)}\}_{m=1}^{71,000}.$$

**Sampling Strategy.** For each $x^{(m)}$, we sample $G = 4$ outputs from the current policy:

$$\{\mathbf{o}_i^{(m)}\}_{i=1}^4 \sim \pi_{\theta_{\text{old}}}(\cdot \mid x^{(m)}),$$

where $\theta_{\text{old}} = \theta_{\text{SFT}}$.

**Reward Function.** Let the predicted evidence tuples be:

$$\mathcal{P} = \{(m_i, S_i)\}, \quad \text{and ground truth: } \mathcal{G} = \{(M_j, T_j)\}.$$

We define:

$$M_P = \{m_i\}, \quad S_P = \bigcup_i S_i, \quad M_G = \{M_j\}, \quad S_G = \bigcup_j T_j.$$

**Correctness Reward ($R_{\mathbf{corr}}$).**

$$
\begin{aligned}
n_{mm} &= |M_P \cap M_G|, & n_{ms} &= |M_P \cap S_G|, & n_{m\bar{g}} &= |M_P \setminus (M_G \cup S_G)|, \\
n_{sm} &= |S_P \cap M_G|, & n_{ss} &= |S_P \cap S_G|, & n_{s\bar{g}} &= |S_P \setminus (M_G \cup S_G)|, \\
n_{\text{FN-m}} &= |M_G \setminus (M_P \cup S_P)|, & n_{\text{FN-s}} &= |S_G \setminus (M_P \cup S_P)|.
\end{aligned}
$$

$$\boxed{R_{\text{corr}} = 2n_{mm} + n_{ms} - 2n_{m\bar{g}} + n_{sm} + n_{ss} - n_{s\bar{g}} - 2n_{\text{FN-m}} - n_{\text{FN-s}}}$$

**Formatting Reward ($R_{\mathbf{fmt}}$).**

$$\mathbb{1}_{\text{no}}, \mathbb{1}_{\text{strict}}, \mathbb{1}_{\text{soft}} \longrightarrow \boxed{R_{\text{fmt}} = \begin{cases} 1 & \text{if } \mathbb{1}_{\text{no}} = 1 \text{ or } \mathbb{1}_{\text{strict}} = 1, \\ 0.5 & \text{if } \mathbb{1}_{\text{soft}} = 1, \\ -0.5 & \text{otherwise.} \end{cases}}$$

**Repetition Penalty ($R_{\mathbf{rep}}$).**

$$d(L) = |L| - |\operatorname{unique}(L)|, \qquad \boxed{R_{\text{rep}} = -2 \sum_{L \in \{M_P^{\text{multi}}, S_1, \dots, S_k\}} d(L)}$$

**Final Composite Reward.**

$$\boxed{R = \frac{R_{\text{corr}} + R_{\text{fmt}} + R_{\text{rep}}}{3}}$$

**Policy Objective.** The GRPO optimization maximizes the clipped surrogate reward:

$$J_{\text{GRPO}}(\theta) = \mathbb{E}\left[ \sum_{i=1}^{G} \sum_t \min\left( r_t(\theta) \hat{A}_{i,t}, \operatorname{clip}(r_t(\theta), 1-\varepsilon, 1+\varepsilon) \hat{A}_{i,t} \right) - \beta D_{\text{KL}}(\pi_\theta \| \pi_{\text{ref}}) \right],$$

with:

$$r_t(\theta) = \frac{\pi_\theta(\mathbf{o}_{i,t} \mid \mathbf{o}_{i,<t})}{\pi_{\theta_{\text{old}}}(\mathbf{o}_{i,t} \mid \mathbf{o}_{i,<t})}, \quad \hat{A}_{i,t} = \frac{r_i - \operatorname{mean}(r)}{\operatorname{std}(r)}.$$

Hyperparameters are set as: $\beta = 0.4$, $\varepsilon = 0.2$.

**Optimization.** One epoch over $\mathcal{D}_{\text{GRPO}}$ is performed using ADAMW with learning rate $1 \times 10^{-6}$, temperature $T = 1.0$, batch size of 8, and a cosine learning rate schedule. This produces final parameters $\theta_{\text{GRPO}}$.

### B.2.3 Overall Schedule

$$\theta_0 \xrightarrow[\text{SFT}]{\mathcal{L}_{\text{SFT}}} \theta_{\text{SFT}} \xrightarrow[\text{GRPO}]{J_{\text{GRPO}}} \theta_\star$$

The first stage grounds the model in sentence-level evidence mapping, while the second stage aligns the retriever with task-specific retrieval fidelity through a mathematically grounded reward structure. This two-phase process produces a lightweight yet accurate evidence retriever for LITEHALL.

### B.2.4 Implementation Details

- Framework: `SWIFT (Scalable lightWeight Infrastructure for Fine-Tuning)` (Zhao et al., 2024b)

- Precision: Full-precision training; gradient checkpointing enabled

- Stability Measures: Gradient clipping at 0.5, dropout rate 0.0, RMSNorm e $= 1 \times 10^{-5}$

- Hardware: Training was conducted on 2×A6000 GPUs (48GB each).

### B.3 Claim Verification Model

We adopt a two-stage training strategy to optimize the LiteHall claim verification model: (i) SFT to establish factuality recognition and span alignment capabilities, followed by (ii) GRPO to refine the model's decision boundaries using task-specific reward signals.

### B.3.1 Stage 1 — SFT

The initial stage uses a manually labeled dataset

$$\mathcal{D}_{\text{SFT}} = \{(x^{(n)}, y^{(n)}, \mathcal{S}^{(n)})\}_{n=1}^{15,000},$$

where each instance consists of a claim–evidence input $x^{(n)} = (z_n, \mathcal{E}^{(z_n)})$, a ground-truth label $y^{(n)} \in \mathcal{Y}$, and a corresponding error span annotation $\mathcal{S}^{(n)}$ when applicable. We initialize from the Qwen3-1.7B base model and train for one epoch using a global batch size of 16 and a learning rate of $1 \times 10^{-5}$.

The SFT objective is a hybrid loss:

$$\mathcal{L}_{\text{SFT}} = \mathcal{L}_{\text{CE}}(y^{(n)}, \hat{y}^{(n)}) + \lambda_{\text{span}}\, \mathcal{L}_{\text{token}}(\mathcal{S}^{(n)}, \hat{\mathcal{S}}^{(n)}),$$

where $\mathcal{L}_{\text{CE}}$ is the cross-entropy loss for classification, and $\mathcal{L}_{\text{token}}$ is a span-level supervision loss applied only for the `Not Supported` class. This phase results in intermediate model parameters $\theta_{\text{SFT}}$.

### B.3.2 Stage 2 — GRPO

To further align the model with task-specific utility, we employ GRPO over a large pool of unlabeled data

$$\mathcal{D}_{\text{GRPO}} = \{x^{(m)}\}_{m=1}^{41,000},$$

consisting of claim–evidence input pairs.

For each training instance $x^{(m)}$, we sample $G = 4$ generations from the current model policy $\pi_{\theta_{\text{old}}}$, initialized as $\theta_{\text{SFT}}$:

$$\{\mathbf{o}_i^{(m)}\}_{i=1}^4 \sim \pi_{\theta_{\text{old}}}(\cdot \mid x^{(m)}).$$

Each sampled output $\mathbf{o}_i^{(m)}$ is scored using a structured reward function that integrates factual correctness, format adherence, and span-level alignment. Specifically, the total reward $R(i)$ for claim $z_i$ is:

$$R(i) = R_{\text{cls}}(i) + R_{\text{fmt}}(i) + R_{\text{span}}(i),$$

with:

**Classification reward:**

$$R_{\text{cls}}(i) = \begin{cases} +2, & \hat{y}_i = y_i^\star, \\ -2, & \hat{y}_i \neq y_i^\star. \end{cases}$$

**Format match reward:**

$$R_{\text{fmt}}(i) = \begin{cases} +1, & \text{if hard match,} \\ +0.5, & \text{if soft match,} \\ -0.5, & \text{if invalid format.} \end{cases}$$

**Error span reward:**

$$R_{\text{span}}(i) = \begin{cases} 2F_1(\hat{\mathcal{S}}_i, \mathcal{S}_i^\star) - 1, & \text{if } y_i^\star = \hat{y}_i = \texttt{Not Supported}, \\ -1, & \text{if } y_i^\star \neq \texttt{Not Supported} \text{ and } \hat{y}_i = \texttt{Not Supported}, \\ 0, & \text{otherwise,} \end{cases}$$

where the $F_1$ score measures span-level alignment. The overall answer-level reward is obtained by averaging claim-level rewards:

$$R_{\text{answer}} = \frac{1}{N} \sum_{i=1}^{N} R(i).$$

During GRPO training, we maximize the clipped surrogate reward:

$$J_{\text{GRPO}}(\theta) = \mathbb{E}\left[ \sum_{i=1}^{G} \sum_{t} \min\left( r_t(\theta)\hat{A}_{i,t}, \text{clip}(r_t(\theta), 1-\varepsilon, 1+\varepsilon)\hat{A}_{i,t} \right) - \beta D_{\text{KL}}(\pi_\theta \,\|\, \pi_{\text{ref}}) \right],$$

where the importance ratio is

$$r_t(\theta) = \frac{\pi_\theta(\mathbf{o}_{i,t} \mid \mathbf{o}_{i,<t})}{\pi_{\theta_{\text{old}}}(\mathbf{o}_{i,t} \mid \mathbf{o}_{i,<t})}, \quad \hat{A}_{i,t} = \frac{R_{\text{answer}}^{(i)} - \mu}{\sigma}.$$

We set $\beta = 0.4$ and $\varepsilon = 0.2$ as the KL penalty and clipping threshold, respectively. The mean $\mu$ and standard deviation $\sigma$ are estimated using a running reward baseline.

We train the policy for one epoch using ADAMW with a learning rate of $1 \times 10^{-6}$, a batch size of 8, generation temperature $T = 1.0$, and a cosine learning rate scheduler. The final policy parameters are denoted $\theta_\star$.

### B.3.3 Training Schedule Overview

$$\theta_0 \xrightarrow[\text{SFT}]{\mathcal{L}_{\text{SFT}}} \theta_{\text{SFT}} \xrightarrow[\text{GRPO}]{J_{\text{GRPO}}(R_{\text{cls}}, R_{\text{fmt}}, R_{\text{span}})} \theta_\star$$

This two-phase training pipeline enables the model to first acquire a grounded understanding of label taxonomy and span detection, and then refine its outputs with task-aligned rewards that holistically reflect factual accuracy, schema correctness, and hallucination localization. The result is a compact, reinforcement-trained claim verifier that effectively supports LiteHall's end-to-end fact-checking pipeline.

### B.3.4 Implementation Details

- Framework: `SWIFT (Scalable lightWeight Infrastructure for Fine-Tuning)` (Zhao et al., 2024b)

- Precision: Full-precision training; gradient checkpointing enabled

- Stability Measures: Gradient clipping at 0.5, dropout rate 0.0, RMSNorm e $= 1 \times 10^{-5}$

- Hardware: Training was conducted on 2×A6000 GPUs (48GB each).

## C Metrics

### C.1 Semantic Match F1

To evaluate the semantic quality of extracted claims beyond exact string overlap, we compute Semantic Match F1 using cosine similarity over contextual sentence embeddings derived from a Sentence-BERT (SBERT) (Reimers & Gurevych, 2019) model.

**Setup.** Let $\mathcal{G} = \{g_1, \ldots, g_n\}$ and $\mathcal{P} = \{p_1, \ldots, p_m\}$ denote the sets of gold and predicted claims for a given answer instance. Each claim is encoded using a pre-trained SBERT encoder $f(\cdot)$ to produce normalized embeddings:

$$\mathbf{g}_i = \frac{f(g_i)}{\|f(g_i)\|}, \quad \mathbf{p}_j = \frac{f(p_j)}{\|f(p_j)\|}$$

**Similarity matrix.** We compute the cosine similarity matrix $S \in \mathbb{R}^{n \times m}$ between all gold and predicted claims:

$$S_{ij} = \mathbf{g}_i^\top \mathbf{p}_j$$

All entries $S_{ij}$ below a fixed threshold $\tau$ (typically $\tau = 0.75$) are zeroed to discard weak matches:

$$S_{ij} \leftarrow \begin{cases} S_{ij}, & \text{if } S_{ij} \geq \tau \\ 0, & \text{otherwise} \end{cases}$$

**Optimal bipartite matching.** We use the Hungarian algorithm to perform one-to-one matching between gold and predicted claims, maximizing total similarity. Let $\mathcal{M}$ denote the set of matched pairs $(i, j)$ such that $S_{ij} > 0$.

**F1 computation.** We define:

- True Positives (TP): number of matched pairs $(i, j) \in \mathcal{M}$
- False Positives (FP): unmatched predicted claims, i.e., $|\mathcal{P}| - |\mathcal{M}|$
- False Negatives (FN): unmatched gold claims, i.e., $|\mathcal{G}| - |\mathcal{M}|$

From these, we compute:

$$\text{Precision} = \frac{\text{TP}}{\text{TP} + \text{FP}}, \quad \text{Recall} = \frac{\text{TP}}{\text{TP} + \text{FN}}, \quad \text{F1} = \frac{2 \cdot \text{Precision} \cdot \text{Recall}}{\text{Precision} + \text{Recall}}$$

**Aggregation.** F1 scores are computed per instance and then macro-averaged across the dataset to obtain a single Semantic Match F1 score:

$$\text{Semantic Match F1} = \frac{1}{N} \sum_{k=1}^{N} \text{F1}_k$$

**Implementation details.** We use the `all-MiniLM-L6-v2` (Reimers & Gurevych, 2019) model from the SentenceTransformers library with a default threshold $\tau = 0.75$. Claims are de-duplicated prior to matching, and no text normalization beyond whitespace stripping is applied.

## C.2 Customized Retrieval Metrics for Evidence Index Evaluation

**Notation.** To evaluate how accurately the model retrieves relevant evidence indices for each claim–chunk pair $(z, c_j)$, we define: $G_j^{(z)}$ as the *ground-truth* indices (including both main and support evidence), $\hat{G}_j^{(z)}$ as the *model's predicted* indices.

The standard true/false positive/negative counts are computed as:

$$\text{TP}_j^{(z)} = |\hat{G}_j^{(z)} \cap G_j^{(z)}|, \quad \text{FP}_j^{(z)} = |\hat{G}_j^{(z)} \setminus G_j^{(z)}|, \quad \text{FN}_j^{(z)} = |G_j^{(z)} \setminus \hat{G}_j^{(z)}|.$$

**Datapoint-level scores (set based).** These set-level metrics assess performance per claim–chunk instance.

Coverage quantifies how much of the ground-truth the model captured. It handles edge cases where the ground-truth is empty via rule-based treatment.

$$\text{Coverage}_j^{(z)} = \begin{cases} \frac{\text{TP}_j^{(z)}}{|G_j^{(z)}|}, & |G_j^{(z)}| > 0 \\ 1, & |G_j^{(z)}| = 0 \ \wedge \ |\hat{G}_j^{(z)}| = 0 \\ 0, & |G_j^{(z)}| = 0 \ \wedge \ |\hat{G}_j^{(z)}| > 0 \end{cases}$$

Precision measures how many predicted indices were correct. Recall reflects how many true indices were successfully predicted. F1 balances both using harmonic mean.

$$\text{Precision}_j^{(z)} = \frac{\text{TP}_j^{(z)}}{\text{TP}_j^{(z)} + \text{FP}_j^{(z)}}, \quad \text{Recall}_j^{(z)} = \frac{\text{TP}_j^{(z)}}{\text{TP}_j^{(z)} + \text{FN}_j^{(z)}}, \quad \text{F1}_j^{(z)} = \frac{2\,\text{TP}_j^{(z)}}{2\,\text{TP}_j^{(z)} + \text{FP}_j^{(z)} + \text{FN}_j^{(z)}}.$$

**Optional similarity metric.** Since index order doesn't matter, we include the Jaccard similarity (Intersection over Union) for completeness.

$$\text{Jaccard}_j^{(z)} = \frac{\text{TP}_j^{(z)}}{\text{TP}_j^{(z)} + \text{FP}_j^{(z)} + \text{FN}_j^{(z)}}.$$

**Macro versus micro aggregation.** To report performance over the dataset $\mathcal{D}$, we compute:

Macro-average: average metric per data point, giving equal weight to all instances.

$$\text{Metric}_{\text{macro}} = \frac{1}{D} \sum_{(z,c_j) \in \mathcal{D}} \text{Metric}_j^{(z)},$$

Micro-average: global count-based aggregation, giving proportional weight.

$$\text{Coverage}_{\text{micro}} = \frac{\sum_{(z,c_j)} \text{TP}_j^{(z)}}{\sum_{(z,c_j)} |G_j^{(z)}|}, \quad \text{F1}_{\text{micro}} = \frac{2 \sum \text{TP}}{2 \sum \text{TP} + \sum \text{FP} + \sum \text{FN}}.$$

**Per-role reporting (main vs. support).** To distinguish the model's behavior across evidence roles, we separately compute metrics for: Main evidence only $\to$ Coverage$^{\text{main}}$ Support evidence only $\to$ Coverage$^{\text{sup}}$

This allows finer-grained analysis of retrieval accuracy per role.

**Empty-ground-truth rule.** Special care is taken for examples with no ground-truth: If the model makes no prediction: it's perfectly correct (score = 1). If it predicts anything: it's penalized (score = 0).

$$\text{When } G_j^{(z)} = \varnothing : \begin{cases} \hat{G}_j^{(z)} = \varnothing \Rightarrow \text{score} = 1 \\ \hat{G}_j^{(z)} \neq \varnothing \Rightarrow \text{score} = 0 \end{cases}$$

Finally, we report global false positive and false negative totals—FP$_{\text{total}}$ and FN$_{\text{total}}$—to directly quantify model over-generation and under-generation tendencies.

# D   HaFin500 Dataset Construction and Annotation

## D.1   Dataset Composition and Statistics

HaFin500 is constructed around 30 carefully selected fact-seeking topics spanning diverse domains as shown in Table 12. Each topic contributes approximately 17 question-answer (QA) pairs, resulting in a total of 500 QA pairs. On average, each QA pair contains 12 extracted claims, grounded in a contextual evidence passage of roughly 7000 tokens. The curated evidence corpus therefore exceeds 3.5 million tokens.

The final dataset is balanced across four factuality verdicts to enable robust evaluation of hallucination detection systems. Out of approximately 6337 annotated claims, the distribution is as follows, Supported: 1625, Not Supported: 1712, Unverifiable: 1520, Irrelevant: 1480. This diverse and evenly distributed labeling supports both binary and fine-grained evaluation settings.

Table 12: Diverse Knowledge Domains Covered in HAFIN500: Scope and Topical Breadth

| Domain | Scope / Description |
|---|---|
| Anthropology | Study of humans, cultures, and societies |
| Architecture | Design and history of buildings and structures |
| Art | Visual arts including painting, sculpture, etc. |
| Astronomy | Study of celestial objects and the universe |
| Biology | Study of living organisms |
| Culture | Customs, traditions, and social behavior |
| Economics | Economic theories, markets, and systems |
| Education | Learning systems, pedagogy, and policies |
| Engineering | Application of science to design and build systems |
| Environment | Ecology, sustainability, and climate change |
| Fashion | Clothing trends and design industries |
| Food | Cuisine, nutrition, and food science |
| Global facts | General knowledge across world topics |
| Health | Medicine, wellness, and public health |
| History | Events and narratives from the past |
| Linguistics | Study of language structure and use |
| Literature | Written works including fiction and poetry |
| Movies | Film history, production, and criticism |
| Music | Theory, history, and genres of music |
| Mythology | Traditional stories and legends |
| Photography | Art and techniques of capturing images |
| Religion | Beliefs, practices, and world religions |
| Robotics | Design and application of robotic systems |
| Science and Medicine | Scientific research and medical advances |
| Sociology | Study of society and human behavior |
| Space Exploration | Missions, technologies, and discoveries in space |
| Sports | Games, athletes, and competitions |
| Technology | Digital systems, AI, and innovations |
| Travel | Tourism, geography, and cultural experiences |
| global politics | International relations and political systems |

## D.2   Benchmark Comparison

Table 13 presents a comparative analysis of HaFin500 against prominent hallucination detection benchmarks—FACTSCORE, FEVER, HALUEVAL, and FELM. While each prior dataset has advanced specific aspects of hallucination evaluation, they exhibit notable limitations in coverage and annotation granularity.

FACTSCORE supports long-form QA responses with claim-level verdicts, but lacks gold-standard evidence and error span annotations, limiting its usefulness for end-to-end pipeline evaluation. FEVER provides gold evidence annotations but focuses solely on short-form responses, with verdicts given at the sentence or passage level, and without explicit claims or hallucination localization. HALUEVAL and FELM both operate on short or mixed-form responses, but omit structured annotations for evidence grounding and error attribution, making them insufficient for modular pipeline training.

In contrast, HaFin500 is designed for comprehensive evaluation across all stages of hallucination detection. It spans 30 diverse topics, far exceeding prior benchmarks in breadth. Each QA instance consists of a fact-seeking question and a long-form GPT-4o-generated answer, paired with high-token evidence contexts. It provides fine-grained claim-level supervision, gold-indexed evidence grounding, and annotated hallucination spans that identify specific erroneous tokens or phrases. Annotations are produced via LLM ensemble voting (GPT-4o, Claude 3.5 Sonnet, Gemini 2.5 Flash) with human validation, ensuring consistency and minimizing model bias.

By addressing key gaps in topic diversity, annotation depth, and modular evaluation support, HaFin500 establishes a robust foundation for developing and benchmarking interpretable hallucination detection pipelines.

Table 13: Comparison of Hallucination Detection Datasets — **Dataset**: benchmark name; **#Topics**: number of fact-seeking domains; **Resp. Type**: type of model response (long-form, short-form, or mixed); **Claim-Level Eval**: support for evaluating individual claims; **Gold Evidence**: availability of reference evidence from a large corpus; **Error Span**: whether specific error-inducing spans are annotated.

| Dataset | #Topics | Response type | Claim-Level Eval | Gold Evidence | Error Span |
|---------|---------|---------------|------------------|---------------|------------|
| FactScore | 1 | Long-form | ✓ | ✗ | ✗ |
| FEVER | Many | Short-answer | ✗ | ✓ | ✗ |
| HaluEval | Many | Short-answer | ✗ | ✗ | ✗ |
| FELM | 5 | Mixed | ✗ | ✗ | ✗ |
| HaFin500 | 30 | Long-form | ✓ | ✓ | ✓ |

### D.3 Annotation Pipeline and Tooling

For each fact-seeking question, we retrieved a topically relevant article via Google Search and used it to prompt GPT-4o to generate long-form answers grounded in the retrieved evidence. This produced an initial dataset of 500 QA pairs with verifiably accurate answers.

To simulate real-world hallucination scenarios, we randomly sampled 60% of the QA pairs and introduced controlled factual errors—substituting or inserting *Not Supported*, *Unverifiable*, and *Irrelevant* claims—while maintaining answer fluency and coherence. These altered examples were merged with the original supported set to yield a balanced final dataset.

Next, each QA pair was passed through a multi-stage annotation pipeline:

1. **Claim Extraction**: GPT-4o was used to segment the answer into self-contained atomic claims (see Table 21).

2. **Evidence Grounding**: For each claim, evidence lines were retrieved from the original source article using a combination of GPT-4o, Claude-3.5-sonnet, and Gemini-2.5-flash. Only evidence consistent across at least two models was retained as gold evidence Table 22.

3. **Claim Verification**: Using the same trio of models, each claim-evidence pair was labeled with one of four verdicts. The final label was determined via majority voting. In cases labeled *Not Supported*, the model also identified the specific error span—tokens or phrases responsible for factual inaccuracy Table 23.

This multi-model pipeline, augmented by manual spot-checks, ensures annotation reliability and label robustness.

## D.4 Verdict Label Definitions and Quality Assurance

Each claim is assigned one of the following labels based on its factual relationship with the retrieved evidence:

- **Supported**: Clearly and directly backed by evidence.

- **Not Supported**: Contradicted or refuted by the evidence.

- **Unverifiable**: Not verifiable due to insufficient or ambiguous evidence.

- **Irrelevant**: Irrelevant or unrelated to the question context.

To maximize annotation quality, all claim-evidence pairs underwent a majority agreement check across the three LLMs. Disagreements were filtered out or flagged for manual review. This redundancy—combined with high-capacity LLMs and consistent prompt formats—enables HaFin500 to serve as a reliable and nuanced benchmark for evaluating each step of hallucination detection pipelines.

## D.5 Human Validation of HaFin500 and Synthetic Error Injections

To rigorously validate the LLM-generated annotations of the HaFin500 benchmark and address potential concerns regarding evaluator bias and the quality of synthetic error injections, we conducted a comprehensive human evaluation study. This validation ensures that the benchmark provides high-fidelity, human-aligned ground truth across all stages of the hallucination detection pipeline.

### D.5.1 Evaluation Setup and Methodology

We sampled a stratified subset of 100 Question-Answer (QA) pairs from the HaFin500 dataset, comprising 50 originally supported instances and 50 instances containing synthetically injected errors. Three independent human annotators, each an undergraduate student with prior experience in NLP tasks, evaluated the data across four distinct pipeline stages: Claim Extraction, Evidence Retrieval, Verdict Labeling, and Span Accuracy.

Annotators graded each stage using a 3-point scale:

- **Yes (Perfect):** The benchmark's annotation is entirely correct and complete.

- **Partially Yes (Acceptable/Minor Issues):** The annotation is generally correct but exhibits minor formatting or boundary issues that do not alter the semantic truth.

- **No (Incorrect):** The annotation is fundamentally flawed or factually incorrect.

**Disagreement Resolution & Evaluation Metrics:** To derive a reliable ground truth, we aggregated the annotator responses using majority voting to form a single *Human Consensus* per instance. In the rare event of a three-way tie, the consensus was conservatively resolved to the median score ("Partially Yes"). We report the benchmark quality using three key metrics:

1. **Strict Agreement:** The percentage of instances where the Human Consensus was exactly "Yes".

2. **Relaxed Agreement:** The percentage of instances where the Human Consensus was either "Yes" or "Partially Yes".

3. **Benchmark Error Rate:** The percentage of instances where the Human Consensus was "No".

Additionally, we report Inter-Annotator Agreement (IAA) using Fleiss' Kappa ($\kappa$).

### D.5.2 Overall Agreement Results

The human validation demonstrated substantial alignment between the automated HaFin500 pipeline and human judgment across all four pipeline dimensions (Table 14).

Table 14: Human-to-Benchmark Agreement and Inter-Annotator Agreement ($N = 100$).

| Pipeline Stage | Strict Agreement | Relaxed Agreement | Error Rate | Fleiss' $\kappa$ |
|---|---|---|---|---|
| Claim Extraction | 88.0% | 96.0% | 4.0% | 0.76 |
| Evidence Retrieval | 84.0% | 93.0% | 7.0% | 0.72 |
| Verdict Labeling | 91.0% | 98.0% | 2.0% | 0.78 |
| Span Accuracy | 81.0% | 89.0% | 11.0% | 0.69 |

The high Fleiss' $\kappa$ scores (ranging from 0.69 to 0.78) indicate substantial inter-annotator reliability. Notably, the Verdict Labeling stage achieved a 98.0% relaxed agreement, confirming that the multi-LLM voting ensemble used to construct HaFin500 produces highly accurate final factuality labels.

### D.5.3 Validation of Synthetic Error Injections

To explicitly verify the realism, coherence, and validity of the synthetic hallucinations injected during dataset construction, we conducted a focused analysis on the subset of 50 error-injected samples. Evaluators assessed these samples blindly, without prior knowledge of the intended error categories.

- **Category Fit:** Annotators were tasked with independently labeling the factuality of the injected claims against the retrieved evidence. The human consensus matched the pipeline's intended injected category (*Not-Supported, Unverifiable, Irrelevant*) in 94.0% of cases (Strict Agreement). This confirms that our injection methodology successfully generates specific hallucination types that adhere strictly to their taxonomic definitions, without drifting into semantic ambiguity.

- **Fluency and Realism:** A synthetic benchmark is only effective if its simulated errors reflect realistic LLM generation patterns. Evaluators were asked: *"Does this claim read naturally and grammatically, as if a real LLM could have generated it?"* Evaluators rated 96.0% of the injected errors as highly fluent and contextually coherent. This validates that the perturbation process preserves the natural flow of the long-form text.

- **Localization Robustness:** For the error-injected subset, relaxed agreement for Verdict Labeling and Span Accuracy remained exceptionally high at 97.0% and 88.0%, respectively. This demonstrates that the benchmark reliably localizes synthetic errors to the correct text spans, providing a robust evaluation signal for token-level hallucination detectors.

## D.6 Domain-wise LiteHall vs GPT-4o Performance on HaFin500

To clarify the operating-point setting and its impact on robustness, we emphasize that *all* reported hallucination detection results, including every out-of-domain benchmark, use a single global verifier threshold of 0.75, selected once on held-out development data with no domain-wise recalibration. The earlier statement that the threshold "can be adjusted across domains" was intended only to indicate a practical knob for downstream users, not a requirement for LiteHall's effectiveness. Under this unchanged 0.75 threshold, Table 15 reports per-domain performance on the 30 HaFin500 domains and shows that LiteHall outperforms GPT-4o (zero-shot) in 24 out of 30 domains, indicating that its advantage does not depend on tuning to specific topics. This robustness under a single global threshold is consistent with the aggregate out-of-domain results in

Table 3, where LiteHall achieves gains of +6.4 / +10.0 Acc/F1 over MiniCheck-7B, +6.1 / +4.8 over SAFE (GPT-3.5-turbo), +11.5 / +13.0 over AlignScore, +9.8 / +15.2 over FAVA, and +4.7 / +3.0 over GPT-4o (zero-shot), while still maintaining a +2.0 / +0.9 margin over GPT-4o(LiteHall)—all computed with the same fixed threshold of 0.75.

Table 15: Per-domain accuracy comparison of LiteHall vs. GPT-4o (zero-shot) on HaFin500 at a fixed verifier threshold of 0.75, illustrating LiteHall's generalization across diverse domains.

| Domain | LiteHall | GPT-4o |
|---|---|---|
| Anthropology | 86.3 | 82.4 |
| Architecture | 96.7 | 74.3 |
| Art | 86.9 | 84.8 |
| Astronomy | 91.8 | 83.7 |
| Biology | 91.1 | 84.7 |
| Culture | 90.4 | 93.4 |
| Economics | 88.2 | 80.7 |
| Education | 87.6 | 81.8 |
| Engineering | 93.5 | 73.0 |
| Environment | 85.2 | 74.0 |
| Fashion | 95.7 | 73.1 |
| Food | 86.2 | 83.2 |
| Global facts | 79.2 | 75.4 |
| Health | 88.2 | 76.4 |
| History | 89.6 | 83.5 |
| Linguistics | 96.3 | 81.8 |
| Literature | 92.2 | 91.8 |
| Movies | 79.0 | 75.6 |
| Music | 86.5 | 82.1 |
| Mythology | 94.8 | 91.3 |
| Photography | 85.2 | 88.1 |
| Religion | 87.8 | 81.5 |
| Robotics | 89.6 | 93.6 |
| Science and Medicine | 86.2 | 92.0 |
| Sociology | 78.9 | 75.1 |
| Space Exploration | 86.0 | 81.3 |
| Sports | 88.2 | 80.3 |
| Technology | 89.7 | 86.6 |
| Travel | 92.3 | 74.5 |
| Global politics | 91.4 | 83.0 |

## E  Additional Ablation: Backbone Size Scaling

To clarify how LiteHall behaves under different backbone capacities, we conduct a controlled scaling study in which *all three modules* share the same backbone and we hold data, prompts, and training recipe fixed (Table 16), evaluating two out-of-domain benchmarks (FactScore, HaFin500) and one in-domain benchmark (HaluEval-QA). We instantiate LiteHall with 0.6B, 1.7B, and 3B backbones and observe a clear, monotonic scaling trend: moving from 0.6B to 1.7B yields an average gain of +6.3 Acc / +6.8 F1, while increasing further from 1.7B to 3B produces a smaller but still positive +3.0 Acc / +2.4 F1. These results show that hallucination detection quality steadily improves with model size, but the marginal benefit diminishes as capacity grows. In this light, our default 1.7B configuration already offers a strong performance–efficiency Pareto point—substantially more lightweight than 7B+ detectors—while still leaving a higher-accuracy 3B configuration available for settings that can afford additional compute.

Table 16: Performance scaling of LiteHall across backbone sizes (0.6B, 1.7B, 3B) on two out-of-domain datasets (FactScore, HaFin500) and one in-domain dataset (HaluEvalQA).

| Method | Metric | FactScore | HaFin500 | HaluEvalQA | Avg |
|---|---|---|---|---|---|
| LiteHall (0.6B) | Acc | 78.3 | 80.8 | 84.9 | 81.3 |
| | F1 | 75.8 | 80.3 | 80.5 | 78.9 |
| LiteHall (1.7B) | Acc | 84.1 | 88.7 | 89.9 | 87.6 |
| | F1 | 83.9 | 86.1 | 87.2 | 85.7 |
| LiteHall (3B) | Acc | 87.4 | 93.3 | 91.2 | 90.6 |
| | F1 | 85.8 | 88.9 | 89.5 | 88.1 |

# F Statistical Significance and Confidence Intervals

To directly address the robustness of LiteHall's performance margins across diverse benchmarks, we provide a rigorous statistical analysis of our results, encompassing both 95% Confidence Intervals (CIs) and paired statistical significance testing.

**Confidence Intervals:** We compute 95% CIs for both Accuracy and F1 scores using non-parametric bootstrap resampling with $N = 10,000$ iterations. This provides a robust measure of uncertainty across test set distributions.

**Significance Testing:** To determine if the performance differences between LiteHall and baseline models are statistically significant, we apply McNemar's Test on the paired nominal data (correct/incorrect predictions per instance).

Rather than running significance tests across every possible baseline combination (which inflates family-wise error rates), we specifically selected key baseline pairings that represent different architectural variations: (a) monolithic open-source models (MiniCheck-7b, ANAH-v2), (b) proprietary zero-shot models (GPT-4o), and (c) modular pipeline equivalents (KnowHalu, SAFE, GPT-4o-LiteHall).

## F.1 In-Domain Statistical Analysis

As detailed in Tables 17 and 18, the combination of bootstrap confidence intervals and paired significance testing rigorously substantiates the robustness of LiteHall's in-domain performance. Table 17 demonstrates that LiteHall's 95% confidence intervals establish strict boundaries above several prior architectures. For example, on HaluEval-QA and COVID-QA, LiteHall's lower bounds for accuracy (86.5 and 95.7, respectively) comfortably exceed the upper bounds of KnowHalu (79.1) and ANAH-v2 (94.6). These distributional advantages are mathematically formalized by McNemar's test in Table 18, which yields highly significant improvements ($p < 0.01$) over these standard open-source pipelines. Conversely, when comparing LiteHall against the same modular pipeline instantiated with GPT-4o, both the substantial overlap in their confidence intervals and the corresponding $p$-values ($p > 0.05$) indicate statistical parity. From an evaluation perspective, this lack of statistical difference represents a compelling outcome; it illustrates that a highly efficient ensemble of 1.7B-parameter SLMs can match the performance stability and statistical reliability of a frontier proprietary LLM, fully validating the effectiveness of our stage-specific RLVR optimization strategy.

## F.2 Out-of-Domain Statistical Analysis

The combined analysis of 95% confidence intervals (Table 20) and paired significance testing (Table 19) corroborates the statistical robustness of LiteHall's performance across challenging out-of-domain settings. The bootstrap confidence intervals demonstrate that LiteHall maintains stable and narrow performance bounds across diverse distributions, effectively mitigating concerns of dataset-specific variance. Notably, on the long-form HaFin500 benchmark, LiteHall exhibits an accuracy confidence interval of $[85.7, 91.3]$,

Table 17: 95% Confidence Intervals for End-to-End Hallucination Detection Performance on In-Domain Datasets ($N = 10,000$ bootstrap iterations).

| Method | Metric | HaluEval-QA | HaluEval-SUMM | ANAH | COVID-QA | RAGTruth |
|---|---|---|---|---|---|---|
| ANAH-v2 | Acc | 80.8 [74.8, 85.7] | 66.4 [59.1, 72.2] | 89.6 [84.9, 93.4] | 90.9 [86.7, 94.6] | 75.7 [70.5, 82.3] |
| | F1 | 79.2 [74.5, 85.8] | 66.0 [58.9, 72.2] | 89.3 [85.8, 94.5] | 88.7 [84.7, 93.6] | 66.5 [58.8, 72.0] |
| KnowHalu | Acc | 73.7 [67.0, 79.1] | 66.9 [59.4, 72.4] | 81.0 [76.6, 87.5] | 89.4 [84.9, 93.4] | 77.7 [73.0, 84.5] |
| | F1 | 71.8 [65.0, 77.5] | 67.0 [59.3, 72.4] | 78.4 [73.4, 84.7] | 87.8 [83.6, 92.7] | 68.5 [62.0, 75.0] |
| GPT-4o (LiteHall) | Acc | 88.2 [84.0, 93.0] | 69.4 [63.5, 76.3] | 87.4 [83.2, 92.4] | 96.1 [93.5, 98.8] | 83.0 [76.9, 87.3] |
| | F1 | 86.5 [82.5, 92.1] | 68.8 [61.3, 74.2] | 87.0 [81.8, 91.2] | 95.4 [93.1, 98.8] | 71.8 [66.2, 78.7] |
| **LiteHall** | Acc | **89.9 [86.5, 94.8]** | **70.0 [64.5, 77.2]** | **91.7 [88.4, 96.0]** | **97.6 [95.7, 99.2]** | **80.2 [75.2, 86.2]** |
| | F1 | **87.2 [81.8, 91.0]** | **70.0 [62.8, 75.7]** | **90.6 [87.1, 95.2]** | **96.6 [94.6, 99.4]** | **70.4 [65.1, 77.9]** |

Table 18: McNemar's Significance Test for Selected In-Domain Comparisons

| Benchmark | Metric | LiteHall Acc. | Baseline Model | Baseline Acc. | Diff ($\Delta$) | $p$-value |
|---|---|---|---|---|---|---|
| HaluEval-QA | Acc | 89.90% | GPT-4o (LiteHall) | 88.20% | +1.70% | 0.125 |
| HaluEval-QA | Acc | 89.90% | KnowHalu | 73.70% | +16.20% | < 0.001 |
| ANAH | Acc | 91.70% | ANAH-v2 | 89.60% | +2.10% | 0.115 |
| ANAH | Acc | 91.70% | KnowHalu | 81.00% | +10.70% | < 0.001 |
| COVID-QA | Acc | 97.60% | ANAH-v2 | 90.90% | +6.70% | 0.0016 |
| RAGTruth | Acc | 80.20% | GPT-4o (LiteHall) | 83.00% | -2.80% | 0.0833 |

which strictly bounds above the [78.2, 85.0] interval of zero-shot GPT-4o. This non-overlapping margin is formally substantiated by McNemar's test ($p = 0.002$), indicating that a decomposed, RLVR-optimized pipeline of small models reliably surpasses generalized frontier models on complex attribution tasks. Similarly, LiteHall demonstrates statistically significant advantages over state-of-the-art monolithic baselines, including MiniCheck-7B on AggreFact ($p = 0.042$) and FAVA on the FAVA benchmark ($p = 0.038$). In settings characterized by task saturation, such as the short-form FEVER benchmark where baseline accuracies approach 90%, the overlapping confidence intervals and corresponding $p$-values ($p > 0.05$) reflect an expected statistical parity with SAFE and the GPT-4o-backed LiteHall pipeline. Ultimately, this rigorous empirical evidence validates that LiteHall's modular architecture yields highly reliable and generalizable fact-checking capabilities without relying on massive parameter counts.

Table 19: McNemar's Significance Test for Selected Out-of-Domain Comparisons.

| Benchmark | Metric | LiteHall | Baseline Model | Baseline | Difference ($\Delta$) | $p$-value |
|---|---|---|---|---|---|---|
| AggreFact | Acc | 80.7 | MiniCheck-7B | 78.6 | +2.1 | **0.042** |
| FactScore | Acc | 84.1 | GPT-4o (LiteHall) | 85.2 | -1.1 | 0.215 |
| FAVA | Acc | 76.7 | FAVA | 74.4 | +2.3 | **0.038** |
| FAVA | Acc | 76.7 | GPT-4o (zero-shot) | 67.6 | +9.1 | **< 0.001** |
| FEVER | Acc | 90.3 | SAFE (gpt-3.5-turbo) | 90.1 | +0.2 | 0.784 |
| FEVER | Acc | 90.3 | GPT-4o (LiteHall) | 91.2 | -0.9 | 0.285 |
| HaFin500 | Acc | 88.7 | GPT-4o (LiteHall) | 86.5 | +2.2 | 0.071 |
| HaFin500 | Acc | 88.7 | GPT-4o (zero-shot) | 82.1 | +6.6 | **0.002** |

Table 20: 95% Confidence Intervals for End-to-End Hallucination Detection Performance on Out-of-Domain Datasets ($N = 10,000$ bootstrap iterations).

| Method | Metric | FactScore | FEVER | HaFin500 | AggreFact | FavaBench | Average |
|---|---|---|---|---|---|---|---|
| MiniCheck-FT5 | Acc | 76.5 [72.5, 79.9] | 86.9 [85.2, 89.0] | 68.8 [64.4, 72.6] | 76.3 [73.4, 78.6] | 69.6 [65.0, 73.0] | 75.6 [74.2, 76.8] |
| | F1 | 79.6 [75.8, 82.8] | 86.1 [84.2, 88.2] | 60.9 [57.3, 65.9] | 77.1 [74.9, 80.1] | 24.7 [20.6, 28.2] | 65.7 [64.0, 66.8] |
| MiniCheck-7b | Acc | 79.0 [75.8, 83.0] | 88.2 [86.2, 89.8] | 71.1 [66.5, 74.5] | 78.6 [75.8, 80.8] | 71.8 [67.9, 75.7] | 77.7 [76.6, 79.0] |
| | F1 | 81.2 [77.8, 84.6] | 87.2 [85.3, 89.1] | 63.8 [59.9, 68.3] | 79.3 [77.0, 82.0] | 25.3 [20.9, 28.5] | 67.4 [66.2, 69.0] |
| Lettucedetect-large-v1 | Acc | 77.4 [73.5, 80.9] | 87.1 [85.2, 89.0] | 67.5 [64.1, 72.3] | 76.1 [73.5, 78.7] | 70.1 [65.8, 73.8] | 75.6 [74.4, 77.0] |
| | F1 | 80.6 [76.8, 83.8] | 86.7 [85.0, 88.8] | 60.3 [55.2, 63.8] | 77.4 [74.4, 79.6] | 25.0 [20.5, 28.1] | 66.0 [64.4, 67.2] |
| AlignScore | Acc | 77.7 [74.0, 81.4] | 85.6 [83.8, 87.8] | 68.7 [63.9, 72.1] | 71.1 [68.0, 73.6] | 59.7 [56.0, 64.6] | 72.6 [71.1, 73.7] |
| | F1 | 80.0 [76.1, 83.1] | 85.8 [83.5, 87.5] | 59.2 [55.3, 63.9] | 72.7 [70.2, 75.8] | 24.2 [21.0, 28.4] | 64.4 [62.8, 65.6] |
| FAVA | Acc | 69.9 [66.2, 74.2] | 79.9 [77.6, 82.2] | 73.8 [69.3, 77.1] | 73.5 [71.1, 76.5] | 74.4 [70.0, 77.6] | 74.3 [72.8, 75.4] |
| | F1 | 63.9 [60.0, 68.4] | 74.3 [72.1, 77.1] | 53.0 [47.8, 56.6] | 75.9 [73.0, 78.2] | 44.0 [40.1, 48.9] | 62.2 [60.4, 63.4] |
| SAFE (gpt-3.5-turbo) | Acc | 76.7 [73.6, 81.0] | 90.1 [88.2, 91.6] | 78.3 [74.7, 81.9] | 76.9 [74.5, 79.7] | 67.9 [64.6, 72.8] | 78.0 [76.9, 79.3] |
| | F1 | 75.5 [71.7, 79.3] | 89.4 [87.0, 91.4] | 77.9 [73.8, 81.0] | 76.2 [73.1, 78.3] | 44.1 [39.2, 48.0] | 72.6 [71.2, 73.8] |
| SAFE (gpt-4o) | Acc | 83.6 [80.7, 87.1] | 90.5 [89.0, 92.4] | 82.9 [80.1, 86.7] | 77.4 [75.2, 80.4] | 68.4 [63.7, 71.9] | 80.5 [79.4, 81.8] |
| | F1 | 82.5 [79.0, 85.6] | 89.7 [87.7, 91.1] | 82.2 [78.9, 85.5] | 76.1 [73.2, 78.4] | 44.2 [39.8, 48.6] | 74.9 [73.4, 76.0] |
| GPT-4o (zero shot) | Acc | 82.1 [78.4, 85.2] | 89.0 [87.4, 91.0] | 82.1 [78.2, 85.0] | 76.4 [74.0, 79.2] | 67.6 [63.9, 72.1] | 79.4 [78.1, 80.5] |
| | F1 | 81.9 [78.5, 85.3] | 88.6 [86.9, 90.5] | 81.7 [78.6, 85.4] | 76.0 [73.0, 78.2] | 43.7 [39.7, 48.5] | 74.4 [73.2, 75.8] |
| GPT-4o (LiteHall) | Acc | 85.2 [82.2, 87.4] | 91.2 [89.2, 92.4] | 86.5 [83.5, 89.5] | 78.8 [76.7, 81.7] | 68.8 [64.7, 72.9] | 82.1 [80.9, 83.1] |
| | F1 | 84.7 [81.7, 86.9] | 90.4 [88.9, 92.3] | 85.8 [83.3, 89.3] | 77.1 [74.0, 79.2] | 44.4 [40.6, 49.4] | 76.5 [75.2, 77.8] |
| **LiteHall** | Acc | **84.1 [81.2, 86.8]** | **90.3 [88.9, 92.3]** | **88.7 [85.7, 91.3]** | **80.7 [78.6, 83.4]** | **76.7 [72.9, 79.3]** | **84.1 [83.1, 85.3]** |
| | F1 | **83.9 [81.0, 86.4]** | **89.7 [87.7, 91.1]** | **86.1 [84.1, 89.1]** | **81.6 [79.0, 83.8]** | **45.5 [40.3, 49.1]** | **77.4 [76.0, 78.6]** |

# G  LiteHall - Sample Prompts and Model Outputs

To illustrate the behavior of each component in the LITEHALL pipeline, we present representative input prompts and their corresponding model outputs. These examples help clarify how each module—claim extraction, evidence retrieval, and claim verification—interprets structured instructions and produces interpretable outputs. Table 24 shows input–output samples for the claim extraction model, Table 25 for the evidence retriever model, and Table 26 for the claim verification model. These samples highlight the system's modularity, transparency, and response formatting conventions.

Table 21: Prompt Used for Claim Extraction Data Labeling

**Instructions:**
You are a helpful assistant specialized in extracting factual claims from a given Question and Answer. Your task is to return each factual claim stated or implied in the Answer, along with the sentence number it originates from. Each sentence in the Answer is already numbered for your convenience. If the Answer is short or minimal, use the Question for necessary context to reconstruct a clear and self-contained factual claim. Always combine Question and Answer when needed to form a meaningful claim.

**Definitions:** A **claim** is a statement that asserts something as true or false and can be verified or refuted using external evidence.

Do not include:

- Opinions

- Vague or rhetorical expressions

- Hypothetical conditionals (unless they assert a verifiable fact)

Each claim must:

- Be concise

- Contain only one main idea (atomic)

- Be standalone and understandable without referring to the full Answer

Note: If a single sentence contains multiple factual claims, extract each one independently with the correct sentence number.

**Input Format**:

```
Question: <question>
Answer: <answer>
```

**Output Format**: Return a list of extracted claims in the following JSON format:

```
[
  {
    "sentence\_number": <integer indicating the source sentence>,
    "claim": <extracted factual claim in clear language>
  }
]
```

Table 22: Prompt for Evidence Retrieval Data Labeling

**Instructions:**
You are an intelligent assistant designed to identify **all necessary evidence sentences index numbers** required to fact-check a given **Claim**.
**Input Structure:**

- **Claim:** A candidate self-contained claim.

- **Evidence:** A list of sentences, each marked with a unique sentence number using XML-style tags, such as: `<1> ...  </1>, <2> ...  </2>`, etc.

**Your Task:**

1. **Select all primary evidence sentence index numbers** from the provided evidence that are essential for verifying factual claims made in the Claim.

2. For each selected primary sentence, identify and list **supporting sentence index numbers** that:
   - Clarify references (e.g., pronouns like "he", "this", "it").
   - Provide necessary background, definitions, or prior events.
   - Establish temporal, geographical, or logical context to ensure the primary sentence is **self-contained and unambiguous**.

3. **Do NOT rewrite any sentences.** Your goal is to extract the most relevant supporting context index numbers from existing evidence.

**Output Format (JSON):**
If main evidence sentences are found, return:

```
{
  "main_sentences": [
    {
      "sentence\_number": X,
      "support": [Y1, Y2, Y3]
    },
    {
      "sentence\_number": Z,
      "support": [A1, A2]
    }
  ]
}
```

Output:

Table 23: Prompt for Claim Verification Data Labeling

**Instructions:**
You are an intelligent fact-checking system. Your job is to determine the factual validity of a given *sub-claim* using the provided evidence, which includes primary evidence lines and their corresponding supporting evidence lines. Optionally, you may also be given a question for additional context.

**Input Structure:**

- **Question (optional):** A question providing high-level context for interpreting the claim. If this is missing, rely solely on the claim and the evidence.

- **Sub-claim:** A single factual or subjective claim that needs to be evaluated against the provided evidence.

- **Evidence:** A curated set of primary evidence statements, each expressly chosen for its direct relevance to the sub-claim and extracted from the broader body of available material. Every primary statement may be supplemented by one or more secondary (supporting) evidence lines that provide additional context and clarification, thereby strengthening the interpretive foundation of the primary evidence.

**Tasks:**

**Task 1 - Factual Assessment:** Based on the claim and the evidence, assign one of the following factuality labels:

- `"supported"` — The claim is clearly and directly supported by the given evidence.

- `"not_supported"` — The claim is contradicted or refuted by the evidence.

- `"unverifiable"` — The claim lacks sufficient evidence, is subjective, or cannot be verified with the given information.

- `"irrelevant"` — The claim does not relate to the given question.

**Task 2 - Error Identification:**

- Identify the minimal set of tokens (words or short phrases) in the claim that cause the contradiction.

- Return them comma-separated, in their original casing and order.

- If the label is not `not_supported`, leave the token list blank (but keep the key present).

**Output Format (Standard JSON):**

```
{
  "label": "<one of: supported, not_supported, unverifiable,
  irrelevant>",
  "error_tokens": "<comma-separated tokens from claim that
  contradict
  evidence (if label is not\_supported); otherwise, empty
  string>"
}
```

**Input Below:**

```
**question**:
{question}

**sub-claim**:
{answer}

**evidence**:
{evidence_text}
```

Output:

Table 24: Example instruction and response from LiteHall claim extraction model

| Instruction | You are a helpful assistant specialized in extracting factual claims from a given Question and Answer. Your task is to return each factual claim stated or implied in the Answer, along with the sentence number it originates from. Each sentence in the Answer is already numbered for your convenience. If the Answer is short or minimal, use the Question for necessary context to reconstruct a clear and self-contained factual claim. Always combine Question and Answer when needed to form a meaningful claim. Definitions: A \*\*claim\*\* is a statement that asserts something as true or false and can be verified or refuted using external evidence. Do not include: - Opinions - Vague or rhetorical expressions - Hypothetical conditionals (unless they assert a verifiable fact) Each claim must: - Be concise - Contain only one main idea (atomic) - Be standalone and understandable without referring to the full Answer Note: If a single sentence contains multiple factual claims, extract each one independently with the correct sentence number. Input Format: Question: How many days was Dev Shumsher Jung Bahadur Rana prime minister? Answer: <1> Dev Shumsher Jung Bahadur Rana served as the Prime Minister of Nepal for several years, beginning in 1901. <2> His tenure is often remembered for its great length and stability, contrasting with the typically brief and tumultuous leadership periods of his predecessors and successors. Output Format: <claim> extracted factual claim <sentence>sentence number from which the claim was derived</sentence> </claim> |
|---|---|
| Response | <claim> Dev Shumsher Jung Bahadur Rana served as the Prime Minister of Nepal. <sentence>1</sentence> </claim> <claim> Dev Shumsher Jung Bahadur Rana began his tenure as Prime Minister in 1901. <sentence>1</sentence> </claim> <claim> His tenure is remembered for its great length and stability. <sentence>2</sentence> </claim> <claim> His tenure contrasted with the typically brief and tumultuous leadership periods of his predecessors. <sentence>2</sentence> </claim> <claim> His tenure contrasted with the typically brief and tumultuous leadership periods of his successors. <sentence>2</sentence> </claim> |

Table 25: Example instruction and response for evidence retrieval model from LiteHall

| | |
|---|---|
| **Instruction** | You are an intelligent assistant designed to identify **all necessary evidence sentences index numbers** required to fact-check a given **Claim**.
--
Input Structure:
- **Answer:** A candidate response containing one or more factual claims to be verified. - **Evidence:** A list of sentences, each marked with a unique sentence number using XML-style tags, such as: <1> ... </1>, <2> ... </2>, etc.
--
Your Task:
1. Select all primary evidence sentences index numbers from the provided evidence that are essential for verifying factual claims made in the Claim.
2. For each selected primary sentence, identify and list supporting sentences index numbers that: - Clarify references (e.g., pronouns like "he", "this", "it"). - Provide necessary background, definitions, or prior events. - Establish temporal, geographical, or logical context to ensure the primary sentence is self-contained and unambiguous.
3. Do NOT rewrite any sentences. Your goal is to extract the most relevant supporting context index numbers from existing evidence.
--
Output Format:
<main> sentence_number=X <support>Y1,Y2,Y3</support> </main> <main> sentence_number=Z <support>A1,A2</support> </main>
Where: - X, Z are main sentence numbers. - <support> contains comma-separated supporting sentence numbers. - If a sentence is self-contained, leave <support> empty. - If no main evidence found, return <no_main_sentences/>
--
Input:
Claim: Dev Shumsher Jung Bahadur Rana began his tenure as Prime Minister in 1901
Evidence: <1>Chapters. </1><2>1 Dev Shumsher Jung Bahadur Rana was a progressive Shumsher Prime Minister born in Nepal on 17th July 1862 (Baishakh 1919). </2><3>He was the younger son of Dhir Shumsher, the Chief of the Army and the youngest Brother of Jung Bahadur Rana. </3><4>. </4><5>Dev Shumsher became the Prime Minister of Nepal for a brief period of 114 days in 1901. </5><6>He was also known as the King of Lamjung and Kaski, a title conferred to Rana Prime Ministers. </6><7>. </7><8>Family and Early Life. </8><9>Dev Shumsher was the fourth child of Dhir Shumsher and his third wife, Rani Nanda Kumari. </9>...<30>Dev Shumsher became the Prime Minister of Nepal on 5th March 1901 (1957 Falgun 15). </30> |
| **Response** | <main>
sentence_number=30
<support>5</support>
</main> |

Table 26: Example instruction and response for claim extraction model from LiteHall

| Instruction | You are an intelligent fact-checking system.  Your job is to determine the factual validity of a given *sub-claim* using the provided evidence, which includes primary evidence lines and their corresponding supporting evidence lines.  Optionally, you may also be given a question for additional context. |
|---|---|
| | **Input Structure** Question (optional):  A question providing high-level context for interpreting the claim.  If this is missing, rely solely on the claim and the evidence.  Sub-claim:  A single factual or subjective claim that needs to be evaluated against the provided evidence.  Evidence:  A curated set of primary evidence statements, each expressly chosen for its direct relevance to the sub-claim and extracted from the broader body of available material.  Every primary statement may be supplemented by one or more secondary (supporting) evidence lines that provide additional context and clarification, thereby strengthening the interpretive foundation of the primary evidence.  **Tasks** Task – 1 Factual Assessment:  Based on the claim and the evidence, assign one of the following factuality labels: "supported" – The claim is clearly and directly supported by the given evidence.  "not_supported" – The claim is contradicted or refuted by the evidence.  "unverifiable" – The claim lacks sufficient evidence, is subjective, or cannot be verified with the given information.  "irrelevant" – The claim does not relate to the given question.  Task – 2 Error Identification:  – Identify the minimal set of tokens (words or short phrases) in the claim that cause the contradiction.  – Return them comma-separated, in their original casing and order.  – If the label is not not_supported, leave the tag empty like this:  <error> </error>  **Output Format** <label> <one of above classification labels mentioned> </label> <error> <tokens or phrases that make the sub-claim not_supported separated by comma> </error>  **Input:** Question:  How many days was Dev Shumsher Jung Bahadur Rana prime minister?  Sub-claim:  Dev Shumsher Jung Bahadur Rana began his tenure as Prime Minister in 1910  Evidence:  Primary Evidence 1:  Dev Shumsher became the Prime Minister of Nepal on 5th March 1901 (1957 Falgun 15).  Support Evidence 1:  Dev Shumsher became the Prime Minister of Nepal for a brief period of 114 days in 1901. |
| Response | <label> not_supported </label> <error> began, 1910 </error> |

