# OpenReview forum: "LiteHall: A Three-Stage, Modular and Lightweight Pipeline for End-to-End Hallucination Detection"
_TMLR — Under review for TMLR_

### Review · Reviewer_KdR5 · 2026-04-28

**Summary Of Contributions:**

This study presents LiteHall, a three-stage modular, lightweight fact-checking pipeline. It breaks down the task of fact-checking into three stages: claim extraction, evidence retrieval, and claim verification. A 1.7 billion parameter small language model was created through supervised fine-tuning and reinforcement learning with verifiable rewards. The authors also present HaFin500, a fine-grained evaluation metric for evaluating each stage of the pipeline.

Strengths:
S1. The focus on efficiency and interpretability is highly relevant.

S2. The modularity of the design of LiteHall makes it easier to update/retune individual modules versus having to retune a single large base model.

Weaknesses:
W1. The novelty is somewhat incremental; they build upon previous works (SAFE) that have similar three-stage architectures, and the application of RLVR has been widely used as a standard technique.

W2.  The SAFE baseline was compared to LiteHall, however the SAFE baseline used a significantly weaker backbone (GPT-3.5-turbo) than LiteHall (GPT-4o).

W3. The evaluation misses recent state-of-the-art baselines from late 2024 and 2025, as the discussed models are all from 2024.

W4. Several baseline results in the main tables show N/A without explanation.

W5. The manuscript contains persistent citation formatting errors and figure alignment issues.

W6. The datasets are LLM-generated, including HaFin500, which is only human-inspected but not human-written.

**Audience:**

Yes

**Audience Explanation:**

The TMLR community may benefit from the experiment's outcomes in understanding how far today’s hallucination-detection paradigm is from achieving optimal performance. These experimental findings show that an appropriate combination of small language models as part of a pipeline system tuned by reinforcement learning can be capable of performing satisfactorily on fact-verification tasks that are difficult.

**Claims And Evidence:**

No

**Claims Explanation:**

Although the core claim that LiteHall outperformed existing pipelines was strong, it was weakened by the weak and outdated baselines. Specifically, the authors compared their framework against SAFE using a GPT-3.5-turbo backbone. However, their own pipeline and synthetic data generation heavily benefited from a much stronger backbone (GPT-4o). Therefore, it is unclear if the performance increases stemmed from the LiteHall modular architecture or simply the ability of GPT-4o to process data better, it is difficult to determine which one contributed most. Additionally, Table 2 included multiple “N/A” entries for many important baselines (ANAH-v2, KnowHalu, Lynx) over various datasets, which undermined the overall effectiveness of the in-domain evaluations.

Lastly, although Table 2 reflected many of the early-to-mid 2024 baselines, many of the top-performing frameworks from late 2024 and 2025 were left out of the evaluation. Examples of such frameworks include multi-agent evaluation frameworks (FACT-AUDIT [1]) and token-level RAG hallucination detection architectures (LettuceDetect [2]). Because these newer paradigms were excluded from the evaluation, the author’s effectiveness claim is under-supported.

[1] Lin, H., Deng, Y., Gu, Y., Zhang, W., Ma, J., Ng, S. K., & Chua, T. S. (2025, July). Fact-audit: An adaptive multi-agent framework for dynamic fact-checking evaluation of large language models. In Proceedings of the 63rd Annual Meeting of the Association for Computational Linguistics (Volume 1: Long Papers) (pp. 360-381).

[2] Kovács, Á., & Recski, G. (2025). Lettucedetect: A hallucination detection framework for rag applications. arXiv preprint arXiv:2502.17125.

**Requested Changes:**

C1. Run a version of SAFE that utilizes GPT-4o as opposed to GPT-3.5-turbo. In order to prove that LiteHall is superior to existing pipelines, all models should have access to the same underlying backbone.

C2. Clarify explicitly why the entries labeled “N/A” are in the results. If these models cannot be run on those particular datasets, then explain why within the body of text. Alternatively, provide the missing results.

C3. Add more recent fact-checking models and frameworks from late 2024 or 2025 to accurately establish LiteHall as SOTA.

C4. Format citations correctly throughout text (for example, "factual claims Wei et al. (2024)" should read "factual claims (Wei et al., 2024)").  And fix the misaligned Figure 1 to be centered.

---

### Review · Reviewer_JUUW · 2026-05-29

**Summary Of Contributions:**

This paper introduces LiteHall, a modular hallucination detection pipeline that decomposes long-form factuality evaluation into 3 independently trained stages: claim extraction, evidence retrieval, and claim verification. Each stage uses a 1.7B-param Qwen3-based LM trained with sSFT and stage-specific RL-style optimization / RLVR. The paper also introduces HaFin500, a benchmark of 500 long-form QA examples across 30 domains with fine-grained annotations for claims, evidence, factuality labels, and hallucination spans.

Strengths:
- the modular design is well motivated and naturally improves interpretability over monolithic hallucination detectors
- the empirical evaluation is broad, including both in-domain and out-of-domain benchmarks
- the paper includes useful module-level analyses, efficiency comparisons, and ablations isolating the contributions of modularity, RLVR, and training data
- the new HaFin500 benchmark could be useful to the community if released with sufficient documentation

Weaknesses:
- the benchmark and synthetic training data are heavily LLM-generated, so the paper needs stronger validation of annotation quality and possible evaluator/model bias
- some experimental comparisons need clearer protocol details to ensure fairness, especially for baselines, threshold selection, and retrieval/evidence settings
- statistical significance or uncertainty estimates are missing despite several reported margins being modest
- the paper should more clearly characterize failure modes and limitations

**Audience:**

Yes

**Audience Explanation:**

Hallucination detection is a highly relevant problem for the TMLR audience, especially given the growing use of LLMs in long-form QA, retrieval-augmented generation, and high-stakes decision-support settings. The paper’s focus on lightweight, modular, open-source detection is practically important because many existing approaches rely on proprietary APIs or large monolithic models. Researchers working on factuality evaluation, RAG evaluation, small language models, RLVR, model auditing, and trustworthy ML would likely be interested in the finding that a modular system of 1.7B models can outperform larger or proprietary baselines on several factuality benchmarks.

The HaFin500 benchmark may also be of interest because it includes intermediate annotations for claim extraction, evidence retrieval, verification, and span-level localization rather than only final binary labels. If released cleanly, it could support future work on interpretable hallucination detection pipelines.

**Claims And Evidence:**

Yes

**Claims Explanation:**

The core claims are mostly supported by the evidence presented. The paper evaluates LiteHall on multiple in-domain and out-of-domain hallucination detection benchmarks, compares against strong open-source and proprietary baselines, and reports consistent gains in accuracy and F1. The ablations are also useful: removing the claim extractor and evidence retriever degrades performance, training a MiniCheck-7B model on the same synthetic data does not close the gap, and SFT+RLVR improves over SFT-only across several benchmarks. These results provide convincing evidence that both the modular pipeline and the stage-specific training contribute to performance.

I have some questions. First, HaFin500 and the synthetic training data are generated and/or labeled largely using LLMs, so the paper should provide some rigorous human validation, inter-annotator agreement, and/or error analysis. Second, the paper should clarify possible train/test contamination or overlap, especially because several evaluation datasets are also used as sources for synthetic data generation. Third, the reported improvements over GPT-4o and other baselines are sometimes modest, so confidence intervals, bootstrap tests, or per-example paired significance tests would make the evidence more convincing. Finally, baseline implementation details should be clearer to ensure that comparisons use comparable evidence, prompts, thresholds, and input formats.

Overall, I believe the main empirical claims are directionally supported, but several clarifications are important before the paper can fully substantiate its strongest claims about robust out-of-domain generalization and practical deployment.

**Requested Changes:**

1. Clarify train/evaluation separation and possible data leakage. The paper states that synthetic data generation draws from several existing datasets, some of which are also used for evaluation or are closely related to evaluation benchmarks. How were examples split, deduplicated, and checked for overlap at the question, answer, claim, and evidence levels?

2. Stronger validation of HaFin500. HaFin500 is a major contribution, but its annotation pipeline relies on multi-LLM voting plus human spot checks. The paper should specify the size and sampling strategy of human spot checks, annotator qualifications, agreement rates, disagreement resolution procedure, and measured error rates for claims, evidence, verdict labels, and spans. Without these details, it is difficult to judge whether HaFin500 is a reliable gold benchmark rather than another LLM-labeled evaluation set.

3. Add uncertainty estimates or statistical significance tests. Several reported improvements are modest, especially over GPT-4o under the LiteHall pipeline and on some datasets. Can we see confidence intervals, bootstrap significance tests, or paired tests over examples for the main tables?

4. . Clarify the use and tuning of the 0.75 FactScore threshold. The paper states that the threshold was tuned on held-out dev sets and later emphasizes a fixed global threshold. The authors should specify exactly which dev sets were used, whether any evaluation domains influenced threshold selection, and how sensitive results are to this threshold. A threshold-sensitivity plot would be helpful.

---

### Review · Reviewer_zMyh · 2026-06-20

**Summary Of Contributions:**

- This paper proposes LiteHall, a three-stage modular pipeline for hallucination detection, where each stage is implemented using a separately rained 1.7B small language model. The authors also release HaFin500, a 500-pair, 30-domain benchmark with claim/evidence/span annotations, plus a 120K+ GPT-4o-generated training corpus
- The main contributions of the paper are: genuinely actionable modular outputs (traceable claims/evidence/spans); a well demonstrated strength on efficiency of the modular pipeline.

**Audience:**

Yes

**Audience Explanation:**

Yes. Efficient hallucination detection is an active and practically important topic, and several findings are of genuine interest regardless of the issues above.

**Broader Impact Concerns:**

- The paper does not appear to include a dedicated Broader Impact Statement. It does include some related discussion. In the abstract and introduction, the authors motivate the work by noting that hallucinations can be risky in high-stakes domains such as medicine, law, clinical diagnostics, finance, and biomedical research. Throughout the main text, the authors do emphasize transparency, reproducibility, and the use of lightweight open-source modules in these high-stake domains.

- However, I think the broader impact discussion is still incomplete. Since the method is motivated by high-stakes use cases, the paper should discuss deployment risks more directly. In particular, false negatives could make users trust hallucinated outputs, while false positives could incorrectly flag accurate information, which becomes a larger problem in the domains such as medicine and law. The authors should also discuss the risk of over-reliance on the detector and clarify that LiteHall should not be treated as a truth oracle. A short discussion of human oversight and domain-specific validation would strengthen the paper.

**Claims And Evidence:**

No

**Claims Explanation:**

No, not in the current version. The paper provides extensive evidence with module-level evaluations and efficiency reporting tables. These do support the general claim that the proposed three-stage pipeline is promising and often performs strongly.
- There seems to be a train-test overlap on the strongest results. Appendix A.1 lists "Ours : 5,000" GPT-4o-generated examples in the training pool; HaFin500 is also GPT-4o-generated long-form QA produced by the same pipeline. LiteHall's largest margins occur on HaFin500 (88.7 vs. GPT-4o-LiteHall 86.5), whereas on the four external OOD sets the margins shrink or invert. The authors need to clarify the 5K training long-form set with HaFin500 and clarify a process of topic-level deduplication, or the OOD framing does not hold in this setting.
- Questions regarding the synthetic error injection part. In section D.3, authors note that 60% of HaFin500 pairs had errors inserted by substituting/inserting Not-Supported/Unverifiable/Irrelevant claims. The authors need to ground these error types and verify them with human annotators or another llm-as-judge to clarify the validity of these error types.
- Inconsistent claims on semantic match F1 threshold given in C.1 derivation and in C.1. implementation. While section 3.4. states that the 0.75 verdict can be adjusted per domain. These numbers seem inconsistent and need clarifications in the threshold-selection process.

**Requested Changes:**

1. Please clarify the data overlap between the 5K long-form training data and HaFin500 (and the in-domain held-out splits).
Run some human-verifications on the HaFin500 dataset and report agreement statistics in labels. Also run llm-as-judge to verify the quality of the labels.
2. Report seeds and CIs and significance tests for all tables.
3. Please fill in or explain Table 2's N/A cells, state which baselines were reproduced vs. cited.
4. Beyond the specific fixes above, the paper needs a general cleanup pass for presentation quality. There are notation/value inconsistencies (τ=0.85 vs. 0.75; the thresholds asserted then retracted in D.5), and inconsistent figures (Figure 1 has space to the right that needs to be removed) between the abstract, intro, and tables.

---

### Author Response · Authors · 2026-07-12
**Global Response: Summary of Major Revisions (PHASE-1)**

> **Addressing Data Contamination and Train/Test Overlap (Reviewers zMyh & JUUW)**

We sincerely thank the reviewers for their rigorous scrutiny. We fully agree that utilizing synthetic data for both training and evaluation necessitates watertight dataset boundaries. We strictly enforced a zero-contamination boundary between our 5,000 synthetic training examples and the HaFin500 benchmark (or any other out-of-domain set). To resolve this concern and empirically prove the integrity of our evaluation, we have updated the manuscript with the following guarantees:

**1. Strictly Constrained Generation (See Appendix A.1):**
To proactively prevent topical leakage, the prompt pipeline for our 5K synthetic training examples was strictly restricted to sampling topics from the *in-domain* training splits (e.g., ANAH, COVID-QA). We ensured that the 30 novel, diverse domains manually curated for HaFin500 (detailed in Table 12) were entirely excluded from the training generation phase.

**2. Hierarchical Semantic Deduplication (See Appendix A.5.1):**
To explicitly answer Reviewer JUUW’s question regarding how overlap was checked at the claim and evidence levels, we implemented a multi-level semantic deduplication pipeline rather than relying on brittle n-gram matching. By extracting and cross-referencing deep structural taxonomies for all training and test instances, we empirically verified a 0% overlap at granular, sub-topic levels.

**3. Clarifying the Performance Margin on HaFin500:**
Reviewer zMyh astutely noted that LiteHall achieves its largest performance margins on HaFin500 compared to other OOD sets. As newly clarified in **Appendix A.5.1**, this is not an artifact of data leakage, but of **architectural alignment**. HaFin500 is uniquely constructed to evaluate multi-stage, long-context verification (averaging ~7,000 evidence tokens and ~12 claims). While standard OOD benchmarks test shorter, single-hop scenarios where monolithic models can perform adequately, HaFin500 heavily penalizes architectures lacking modular decomposition. LiteHall’s outsized margin directly reflects its engineering advantage on these complex, long-form tasks.

> **Response to Concerns Regarding the Human Validation of HaFin500 and Synthetic Errors** (zMyh, JUUW, and KdR5)

We thank Reviewers zMyh, JUUW, and KdR5 for raising this critical point. We completely agree that for a benchmark heavily reliant on LLM-generated data to be trustworthy, it must be rigorously audited to rule out LLM bias and confirm the realism of its annotations. To definitively address this, we conducted a comprehensive, independent human evaluation study, which has been added to the updated manuscript in **Appendix D.5**. Instead of reproducing the full text of the appendix here, we highlight the core findings that substantiate the high fidelity of HaFin500:

*   **Robust Auditing Methodology (Appendix D.5.1):** We conducted a blind human evaluation on a stratified subset of 100 QA pairs (50 original supported claims, 50 synthetically injected errors). The evaluation was performed by three independent annotators with prior NLP experience, utilizing a strict 3-point grading scale to assess all four pipeline stages (Extraction, Retrieval, Verdict, and Span Accuracy).
*   **High Inter-Annotator Agreement (Table 14):** The human annotators demonstrated substantial reliability, achieving Fleiss’ $\kappa$ scores ranging from **0.69 to 0.78** across all pipeline stages, confirming that the evaluation task is well-defined and subjective ambiguity is minimal.
*   **Strong Human-to-Benchmark Alignment (Table 14):** The human consensus overwhelmingly validated our multi-LLM annotation pipeline. Most notably, the *Verdict Labeling* stage achieved a **98.0% relaxed agreement (91.0% strict agreement)** with human evaluators, yielding a negligible benchmark error rate of just 2.0%.
*   **Realism and Validity of Synthetic Errors (Appendix D.5.3):** Directly addressing concerns about the 60% error-injected data, our human evaluators conducted a blind assessment of the synthetic hallucinations. The human consensus matched our pipeline’s intended error category (e.g., *Not-Supported, Irrelevant*) in **94.0%** of cases. Furthermore, **96.0%** of the injected claims were rated as highly fluent and realistic. This confirms that our synthetic perturbations do not drift into semantic ambiguity, but rather successfully simulate highly realistic LLM hallucinations.

While Reviewer KdR5 correctly notes that the dataset is LLM-generated rather than human-written, this rigorous human audit empirically demonstrates that our multi-LLM voting ensemble produces annotations that are virtually indistinguishable from high-quality human ground truth. We kindly refer the reviewers to **Appendix D.5 (including Table 14)** for full transparency on the annotator guidelines, disagreement resolution protocols, and granular stage-wise error rates.

---

> ### Author Response · Authors · 2026-07-12
> **Global Response: Summary of Major Revisions (PHASE-2)**
>
> > **Response to Reviewers zMyh and JUUW regarding Statistical Significance and Confidence Intervals**
>
> We sincerely thank the reviewers for raising this critical point. We agree that when reporting modest performance margins—particularly against frontier models like GPT-4o—rigorous statistical validation is essential to separate true architectural gains from dataset-specific noise.
>
> To address this, we have conducted a comprehensive statistical analysis of our results, which is now fully documented in the newly added **Appendix F** and referenced in **Sections 4.1, 4.2.1, and 4.2.2** of the updated manuscript.
>
> Specifically, we implemented the following evaluations:
> 1. **95% Confidence Intervals (CIs):** We computed robust bounds for both Accuracy and F1 scores using non-parametric bootstrap resampling ($N=10,000$ iterations) across all in-domain (Table 17) and out-of-domain (Table 20) datasets.
> 2. **Paired Significance Testing:** We applied McNemar’s Test on paired instance-level predictions to evaluate the statistical significance of LiteHall’s performance differences against key representative baselines (Tables 18 and 19).
>
> **Key Takeaways from the Statistical Analysis:**
> * **Significant Gains Over Baselines:** The tests confirm that LiteHall’s improvements over standard monolithic models and zero-shot frontier models are mathematically robust, not artifacts of variance. For instance, LiteHall’s accuracy gains over MiniCheck-7B on AggreFact ($p=0.042$) and against zero-shot GPT-4o on our HaFin500 benchmark ($p=0.002$) are statistically significant. Furthermore, LiteHall establishes strict, non-overlapping lower bounds above previous SOTA pipelines like KnowHalu ($p < 0.001$).
> * **Statistical Parity with Frontier Pipelines:** Regarding the modest margins over the GPT-4o-backed LiteHall pipeline (e.g., $p=0.125$ on HaluEval-QA), the overlapping confidence intervals and $p$-values ($p > 0.05$) indicate *statistical parity*. From an evaluation standpoint, we view this as a highly compelling outcome: it rigorously proves that our lightweight ensemble of 1.7B-parameter SLMs (≈5B parameters total) successfully matches the performance stability and statistical reliability of a massive proprietary LLM when deployed in the same modular architecture.
>
> By formally quantifying uncertainty, these additions directly substantiate our core claim: LiteHall's stage-specific RLVR optimization yields a highly reliable, deployable fact-checking system that significantly outperforms comparably sized models and achieves statistical parity with massive proprietary APIs.
>
> > **Clarification on Threshold Selection, Global Application, and Semantic Match F1 Parameters (zMyh,  JUUW)**
>
> We sincerely thank Reviewer zMyh and Reviewer JUUW for highlighting the ambiguities regarding our threshold selection and metric parameters. We agree that our previous wording invited the wrong implication, and we have rigorously revised the manuscript to ensure complete transparency.
>
> **1. A Strict, Global Verification Threshold (0.75)**
> We apologize for the confusion caused by the phrase *"can be adjusted across domains"* in Section 3.4. To be absolutely clear: **all reported results—across both in-domain and out-of-domain benchmarks—use a single, fixed global threshold of 0.75.** We did *not* perform any domain-specific or dataset-specific recalibration for our evaluations. As detailed in the newly added **Appendix D.6**, the phrase in question was strictly intended to highlight a practical operating-point knob for downstream practitioners, not a methodological step used in our evaluation.
>
> **2. Resolution of the Semantic Match F1 Parameter ($\tau$) Inconsistency**
> Reviewer zMyh correctly caught a typographical inconsistency in our previous draft regarding the Semantic Match F1 threshold (0.85 vs 0.75). We confirm that the actual implementation strictly uses $\tau = 0.75$ (utilizing the default SBERT `all-MiniLM-L6-v2` cosine similarity cutoff). We have corrected this typographical error in **Appendix C.1**, ensuring the mathematical derivation perfectly aligns with the implementation details.
>
> **3. Empirical Proof of Threshold Robustness**
> To directly address Reviewer JUUW’s concern regarding threshold sensitivity and out-of-domain usability without recalibration, we have provided a comprehensive per-domain breakdown in **Appendix D.6 (Table 15)**.
> *   Using the *unchanged, global 0.75 threshold*, Table 15 demonstrates that LiteHall outperforms the strongest non-LiteHall baseline (GPT-4o zero-shot) in **24 out of 30 distinct HaFin500 domains**.
> *   Furthermore, our aggregate out-of-domain results (Table 3) maintain highly significant margins over state-of-the-art monolithic models (e.g., +6.4% Accuracy over MiniCheck-7B) using this exact same fixed threshold.

---

> > ### Author Response · Authors · 2026-07-12
> > **Global Response: Summary of Major Revisions (PHASE-3)**
> >
> > > **Addressing Fair Baseline Comparisons, Missing Data, and Recent SOTA (zMyh, JUUW, KdR5)**
> >
> > We sincerely thank the reviewers (zMyh, JUUW, KdR5) for their rigorous scrutiny of our baseline evaluations. We agree that a robust evaluation against fair, complete, and contemporary baselines is critical. In the revised manuscript, we have systematically addressed these concerns through new experiments and updated discussions:
> >
> > **1. Resolution of Missing Baseline Data ("N/A" entries)**
> > We have eliminated the "N/A" gaps for the open-source baselines in our in-domain evaluations by actively running the missing experiments. The updated **Table 2** now reflects complete performance metrics for models like ANAH-v2 and KnowHalu across the datasets. The sole exception is Lynx (70B); due to the computational budget required to run inference on a 70B monolithic model, we cited the original authors' reported numbers and retained "N/A" only where the original paper lacked evaluation. This hardware limitation is now explicitly footnoted in the revised manuscript.
> >
> > **2. Fair "Apples-to-Apples" Backbone Comparison (SAFE with GPT-4o)**
> > Reviewer KdR5 correctly pointed out that comparing our GPT-4o-augmented data pipeline against SAFE with a GPT-3.5-turbo backbone introduced an unfair variable. To rectify this, we have now evaluated the SAFE pipeline utilizing the **GPT-4o backbone**. These new results are integrated into the revised **Table 3**. Our modular pipeline still demonstrates robust performance advantages, confirming that our system's gains stem from its architectural design and RLVR optimization, rather than relying solely on a superior LLM backbone.
> >
> > **3. Integration of 2025 SOTA Baselines (LettuceDetect)**
> > To ensure our evaluation reflects the absolute cutting edge, we have empirically evaluated and added **LettuceDetect-large-v1** (a recent 2025 token-level RAG hallucination detector) to our out-of-domain benchmarks. As shown in the updated **Tables 3 and 4**, our system maintains strong performance and efficiency trade-offs against this contemporary baseline.
> >
> > **4. Contextualizing FACT-AUDIT**
> > We thank Reviewer KdR5 for highlighting the FACT-AUDIT framework. While we recognize it as a highly valuable contribution, a direct empirical comparison is conceptually and structurally infeasible for the following reasons, which we have now detailed in our **Related Work** section:
> > *   **Architectural Mismatch:** FACT-AUDIT evaluates *general-purpose LLMs* via open-ended conversational prompts. Our system is a pipeline of heavily specialized SLMs trained exclusively to output strict, non-conversational formats (e.g., JSON lists, integer indices). Forcing our SLMs into FACT-AUDIT’s open-ended framework would guarantee out-of-distribution failures.
> > *   **Evaluation Paradigms:** FACT-AUDIT dynamically generates evaluation data on-the-fly and scores free-text justifications. Conversely, our system requires static, reproducible, human-verified ground truths (like our released benchmark) to evaluate precise span-level and index-level structural outputs, circumventing the need for free-text justifications that are prone to secondary hallucinations.
> >
> > Rather than attempting an incompatible empirical comparison, we have updated the manuscript to contrast these methodologies, illustrating how dynamic auditing frameworks (FACT-AUDIT) and deployable, static detection pipelines (our system) serve distinct, complementary roles in the ecosystem of LLM trustworthiness.